# CERTIFY OR PREDICT: BOOSTING CERTIFIED ROBUSTNESS WITH COMPOSITIONAL ARCHITECTURES

**Mark Niklas Müller, Mislav Balunović, Martin Vechev**
Department of Computer Science, ETH Zurich, Switzerland
{mark.mueller, mislav.balunovic, martin.vechev}@inf.ethz.ch

## ABSTRACT

A core challenge with existing certified defense mechanisms is that while they improve certified robustness, they also tend to drastically decrease natural accuracy, making it difficult to use these methods in practice. In this work, we propose a new architecture which addresses this challenge and enables one to boost the certified robustness of any state-of-the-art deep network, while controlling the overall accuracy loss, without requiring retraining. The key idea is to combine this model with a (smaller) certified network where at inference time, an adaptive selection mechanism decides on the network used to process the input sample. The approach is compositional: one can combine any pair of state-of-the-art (e.g., EfficientNet or ResNet) and certified networks, without restriction. The resulting architecture enables much higher natural accuracy than previously possible with certified defenses alone, while substantially boosting the certified robustness of deep networks. We demonstrate the effectiveness of this adaptive approach on a variety of datasets and architectures. For instance, on CIFAR-10 with an $\ell_\infty$ perturbation of 2/255, we are the first to obtain a high natural accuracy (90.1%) with non-trivial certified robustness (27.5%). Notably, prior state-of-the-art methods incur a substantial drop in accuracy for a similar certified robustness.

## 1 INTRODUCTION

Most recent defenses against adversarial examples have been broken by stronger and more adaptive attacks (Athalye et al., 2018; Tramer et al., 2020), highlighting the importance of investigating certified defenses with suitable robustness guarantees (Raghunathan et al., 2018; Wong & Kolter, 2018; Zhang et al., 2020; Balunović & Vechev, 2020). And while there has been much progress in developing new certified defenses, a fundamental roadblock to their practical adoption is that they tend to produce networks with an unsatisfying natural accuracy.

In this work we propose a novel architecture which brings certified defenses closer to practical use: the architecture enables boosting certified robustness of state-of-the-art deep neural networks without incurring significant accuracy loss and without requiring retraining. Our proposed architecture is compositional and consists of three components: (i) a *core-network* with high natural accuracy, (ii) a *certification-network* with high certifiable robustness (need not have high accuracy), and (iii) a *selection mechanism* that adaptively decides which one of the two networks should process the input sample. The benefit of this architecture is that we can plug in any state-of-the-art deep neural network as a core-network and any certified defense for the certification-network, thus benefiting from any future advances in standard training and certified defenses.

A key challenge with certifying the robustness of a decision made by the composed architecture is obtaining a certifiable selection mechanism. Towards that, we propose two different selection mechanisms, one based on an auxiliary selection-network and another based on entropy, and design effective ways to certify both. Experimentally, we demonstrate the promise of this architecture: we are able to train a model with much higher natural accuracy than models trained using prior certified defenses while obtaining non-trivial certified robustness. For example, on the challenging CIFAR-10 dataset with an $\ell_\infty$ perturbation of 2/255, we obtain 90.1% natural accuracy and a certified robustness of 27.5%. On the same task, prior approaches cannot obtain the same natural accuracies for any non-trivial certified robustness.

**Main contributions**   Our main contributions are:

- A new architecture, called ACE (short for Architecture for Certification), which boosts certified robustness of networks with high natural accuracy (e.g., EfficientNet).

- Methods to train our newly proposed architecture and to certify the robustness of the entire composed network, including the certification of the selection mechanism.

- Experimental evaluation on the CIFAR-10, TinyImageNet and ImageNet200 datasets, demonstrating the promise of ACE: at the same non-trivial certified robustness levels, we can achieve significantly higher accuracies than prior work.

- We release our code as open source: `https://github.com/eth-sri/ACE`

## 2   RELATED WORK

There has been much recent work on certified defenses, that is, training neural networks with provable robustness guarantees. These works include methods based on semidefinite relaxations (Raghunathan et al., 2018), linear relaxations and duality (Wong & Kolter, 2018; Wong et al., 2018; Xu et al., 2020), abstract interpretation (Mirman et al., 2018), and interval bound propagation (Gowal et al., 2018). The three most recent advances are COLT (Balunović & Vechev, 2020), based on convex layer-wise adversarial training, CROWN-IBP (Zhang et al., 2020), based on a combination of linear relaxations Zhang et al. (2018) and interval propagation, and LiRPA (Xu et al., 2020) scaling to problems with many more classes by directly bounding the cross entropy loss instead of logit margins. As mentioned earlier, a key challenge with these methods is that in order to gain certified robustness, they tend to incur a drastic drop in natural accuracy.

In parallel to certified defenses, there has also been interest in certifying already trained models (Katz et al., 2017; Tjeng et al., 2017; Gehr et al., 2018; Weng et al., 2018; Bunel et al., 2018; Wang et al., 2018a; Singh et al., 2019). While these methods were initially focused mostly on $L_p$ robustness, these works (as well as ours) can be naturally extended to other notions of robustness, such as geometric (Balunović et al., 2019) or semantic (Mohapatra et al., 2020) perturbations. A line of work that weakens deterministic guarantees so to scale to larger networks is that of randomized smoothing which offers probabilistic guarantees (Lecuyer et al., 2018; Cohen et al., 2019; Salman et al., 2019a). While interesting, this technique incurs overhead at inference time due to additional sampling, and further, generalizing smoothing to richer transformations (e.g., geometric) is non-trivial (Fischer et al., 2020). In contrast, our work handles large networks while providing deterministic guarantees and because of its compositional nature, directly benefits from any advancements in certification and certified defenses with richer perturbations.

Our proposed architecture is partially inspired by prior work on designing custom architectures for dynamic routing in neural networks (Teerapittayanon et al., 2016; Bolukbasi et al., 2017; McGill & Perona, 2017; Wang et al., 2018b). While the main goal of these architectures is to speed up inference, our observation is that similar type of ideas are applicable to the problem of enhancing certifiable robustness of existing neural networks.

## 3   BACKGROUND

We now present the necessary background needed to define our method.

**Adversarial Robustness**   We define adversarial robustness of a model $h$ as a requirement that $h$ classifies all inputs in a $p$-norm ball $B_\epsilon^p(\boldsymbol{x})$ of radius $\epsilon$ around the sample $\boldsymbol{x}$ to the same class:

$$\arg\max_j h(\boldsymbol{x})_j = \arg\max_j h(\boldsymbol{x}')_j, \quad \forall \boldsymbol{x}' \in B_\epsilon^p(\boldsymbol{x}) := \{\boldsymbol{x}' = \boldsymbol{x} + \boldsymbol{\eta} \mid \ |\boldsymbol{\eta}|_p \leq \epsilon_p\} \quad (1)$$

In this work we focus on an $\ell_\infty$ based threat model and use the notation $\epsilon_p$ to indicate the upper bound to the $\ell_p$-norm of admissible perturbations. The robust accuracy of a network is derived from this definition as the probability that an unperturbed sample from the test distribution is classified correctly and Equation 1 holds. As it is usually infeasible to compute exact robustness, we define certifiably robust accuracy (also certifiable accuracy or certifiable robustness), as a provable lower

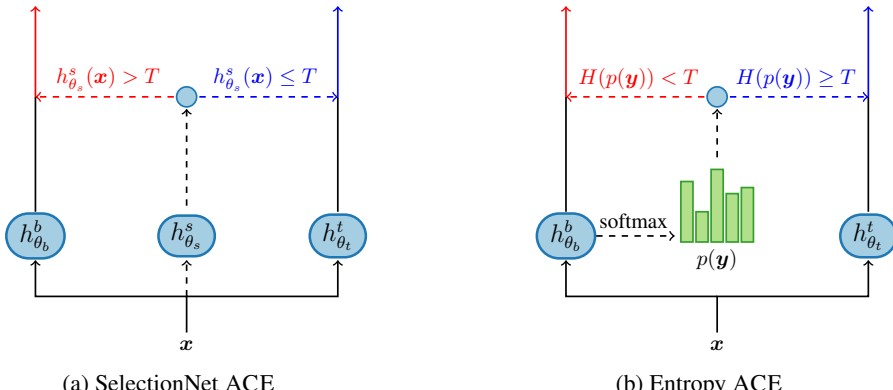

(a) SelectionNet ACE           (b) Entropy ACE

Figure 1: Two variants of ACE. Dashed and solid arrows represent selection mechanism and classification networks, respectively. The colored, dashed arrows represent the selection decision obtained by thresholding either the output of the selection network or the entropy. Based on this selection, we output either the result of the certification-network $h_{\theta_b}^b$ (red) or the core-network $h_{\theta_t}^t$ (blue).

bound to the robust accuracy of a network. This lower bound is obtained by attempting to prove adversarial robustness for all correctly classified samples $\boldsymbol{x}$ using any certification method. For a fixed, arbitrary certification method, we introduce the binary function $\mathrm{cert}(\mathbb{X}, f, y)$ as the result of an attempt to certify that $f(\boldsymbol{x}) = y, \ \forall \boldsymbol{x} \in \mathbb{X}$. In practice, adversarial robustness is usually evaluated as an upper bound to the exact robust accuracy of a network and denoted as adversarial accuracy. This upper bound is usually computed using an adversarial attack such as PGD (Madry et al., 2017).

**Certification and Training with Convex Relaxations**    Here we briefly summarize robustness certification via convex relaxations. The idea is to start with an initial convex set capturing all admissible perturbations of an original sample $\boldsymbol{x}$, denoted as $\mathbb{C}_0 \supseteq B_\epsilon^\infty(\boldsymbol{x})$ and then propagate this convex set sequentially through all layers of the network. The key challenge is to design a transformer $T_f$ that maps a convex input set to a convex output set for every function $f$ corresponding to a network layer, while ensuring *soundness*: we need to guarantee that each point $\boldsymbol{x}$ in the convex input set $\mathbb{C}_{in}$ is mapped to a point in the convex output set, i.e. $f(\boldsymbol{x}) \in \mathbb{C}_{out} := T_f(\mathbb{C}_{in})$. Finally, we obtain a convex shape that captures all possible outputs of the network. To prove robustness, we have to show that the output of the target class is greater than that of any other class, which is often simple. Depending on the type of transformer $T_f$ used, this framework leads to various certification methods such as IBP (Gowal et al., 2018) or DeepZ (Singh et al., 2018). A more comprehensive description of these methods can be found in Salman et al. (2019b). There has recently been a plethora of work which uses these methods to train provably robust neural networks, which we reference in Section 2. In this work we use models pretrained with CROWN-IBP (Zhang et al., 2020), LiRPA (Xu et al., 2020) and COLT (Balunović & Vechev, 2020) and train selection- and certification-networks using CROWN-IBP, IBP and COLT. IBP (Mirman et al., 2018; Gowal et al., 2018) and CRONW-IBP compute convex relaxations of the loss using intervals and restricted polyhedra, respectively, and minimize this loss during training. COLT (Balunović & Vechev, 2020) proceeds layer by layer and tries to find adversarial examples inside the convex relaxations of the latent spaces of perturbed samples, using PGD adversarial attacks.

## 4   COMPOSITIONAL ARCHITECTURE FOR CERTIFIABLE ROBUSTNESS

In this section, we formally introduce our proposed compositional architecture.

**Overview**    We first describe the idea behind the two variants of our architecture, illustrated in Figure 1. The first variant, named SelectionNet ACE, is shown in Figure 1a. Here, the selection mechanism is an auxilliary selection network $h_{\theta_s}^s$ which decides whether to pass sample $\boldsymbol{x}$ through the core- or the certification-network. If the output of the selection network is greater than $T$, then we pass $\boldsymbol{x}$ through the certification-network $h_{\theta_b}^b$, and otherwise we pass it through the core-network $h_{\theta_t}^t$. The second variant, Entropy ACE, is shown in Figure 1b. Here we perform the selection for

every sample $\boldsymbol{x}$ based on the output probabilities $p(\boldsymbol{y})$ produced by the certification-network $h_{\theta_b}^b$ via the softmax function. If the entropy $H(p(\boldsymbol{y}))$ of the output probability distribution exceeds a fixed threshold $T$, we pass the sample through the core-network $h_{\theta_t}^t$, otherwise we return the output of the certification-network. In Figure 1 we show the selection mechanism using dashed arrows and the two possible outputs of the end-to-end architecture using solid, red and blue arrows.

Formally, we propose the compositional neural network architecture $h_\theta : \mathcal{X} \to \mathcal{Y}$ defined as

$$h_\theta(\boldsymbol{x}) = g_{\theta_s}(\boldsymbol{x}) \cdot h_{\theta_b}^b(\boldsymbol{x}) + (1 - g_{\theta_s}(\boldsymbol{x})) \cdot h_{\theta_t}^t(\boldsymbol{x}) \tag{2}$$

The architecture combines three components: A selection-mechanism $g_{\theta_s} : \mathcal{X} \to \{0, 1\}$ decides whether to forward an input $\boldsymbol{x}$ through the core- or certification-network ($g_{\theta_s}$ is instantiated in two different ways below), while the core-network $h_{\theta_t}^t : \mathcal{X} \to \mathcal{Y}$ and the certification-network $h_{\theta_b}^b : \mathcal{X} \to \mathcal{Y}$ assign an output label $y \in \mathcal{Y}$ to an input $\boldsymbol{x} \in \mathcal{X}$. We note that arbitrary network architectures and training methods can be used for each of the component networks. We evaluate some of these choices in our experimental evaluation section later.

## 4.1 Selection Mechanism

The core of ACE is a selection mechanism that decides which network to use for inference. Ideally, the selection mechanism should pass inputs for which the certification-network is correct and certifiably robust through the certification-network, and all other inputs through the core-network. To train this mechanism, we set the following selection target for each sample $\boldsymbol{x}$ in the training set:

$$y_s(h_{\theta_b}^b, \boldsymbol{x}) = \mathrm{cert}(B_\epsilon^\infty(\boldsymbol{x}), h_{\theta_b}^b, y) \tag{3}$$

The output of the binary function $\mathrm{cert}$, explained in Section 3, is 1 if and only if we can certify that network $h_{\theta_b}^b$ classifies all inputs from the region $B_\epsilon^\infty(\boldsymbol{x})$ to a label $y$.

If the separation described above were fully accurate and certifiable, the certifiable robustness of the certification-network would be retained by the combined network, while the natural accuracy and adversarial robustness would be lower bounded by those of the core-network. However, as the task of predicting certifiable correctness is a strictly more difficult variant of the meta recognition task (Scheirer et al., 2012), a perfect selection is usually unattainable. Here, we balance a trade-off between certifiable and natural accuracy, as a higher selection rate and consequently recall will generally increase certifiable accuracy at the cost of a larger drop in natural and adversarial accuracy. Clearly, a lower selection rate and consequently higher precision have the opposite effect. Note that, if there exist perturbations $\boldsymbol{x}_1', \boldsymbol{x}_2' \in B_\epsilon^\infty(\boldsymbol{x})$ such that $g_{\theta_s}(\boldsymbol{x}_1') = 0$ and $g_{\theta_s}(\boldsymbol{x}_2') = 1$, then we would have to certify the robustness of both the core- and certification-network (we would like to avoid certifying the core-network). Thus, only if we can prove that such a pair does not exist, meaning that the selection is robust, we can certify only one of these two networks.

As the typically large and deep core-network tends to have a very low certifiable accuracy, low robustness of the selection mechanism leads directly to low certifiable accuracy of the composed network (because we would need to certify the core-network more often). An ideal selection mechanism has to be robust, accurate, and allow tuning of the selection rate.

Towards that, we suggest two variants that fit these requirements: (i) SelectionNet, a selection-network trained on the binary selection task, and (ii) Entropy Selection, based on a threshold on the entropy of the output of the certification-network. We refer to the resulting architectures as SelectionNet ACE and Entropy ACE.

### 4.1.1 SelectionNet

We propose using a selection-network $h_{\theta_s}^s : \mathcal{X} \to \mathbb{R}$ to make the decision, which leads to the architecture illustrated in Figure 1a with the following inference procedure: (i) Pass sample $\boldsymbol{x}$ through the selection-network resulting in the output $h_{\theta_s}^s(\boldsymbol{x})$. (ii) If the output is greater than the threshold $T$, the sample is passed through the certification-network, and otherwise through the core-network. Formally, we define the selection mechanism as $g_{\theta_s}(\boldsymbol{x}) := \mathbb{1}_{h_{\theta_s}^s(\boldsymbol{x}) > T}$.

**Training** After training a provable certification-network using any choice of a provable training mechanism, we obtain the selection targets according to Equation 3. We then frame the selection

problem as a binary classification task and train the selection-network directly on the inputs from the training set using the selection targets obtained above as labels. We note that for the selection-network, similarly to the certification-network, one can use any network architecture and provable training mechanism. Further, we propose to reduce the training time by using the certification-network as a feature extractor and only training the last linear layer of the selection-network. This also reduces the selection overhead during inference and certification with convex relaxation based methods. The core-network is trained completely independently, typically using a pre-trained model.

### 4.1.2 ENTROPY SELECTION

As a second variant we propose using an entropy based selection mechanism $g_{\theta_s}(\boldsymbol{x}) := H(\text{softmax}(h_{\theta_b}^b(\boldsymbol{x}))) < T$, inspired by Teerapittayanon et al. (2016), which leads to the network architecture illustrated in Figure 1b and a slightly different inference procedure:

We first map the output of the certification-network $\boldsymbol{y}_b = h_{\theta_b}^b(\boldsymbol{x})$ to a discrete probability distribution $p(\boldsymbol{y}_b)$ over the labels, via the softmax function. Then we can compute the entropy $H(p(\boldsymbol{y}_b))$ as

$$H(p(\boldsymbol{y}_b)) = -\sum_{j=1}^{n} p(\boldsymbol{y}_b)_j \log(p(\boldsymbol{y}_b)_j). \tag{4}$$

If the entropy is below a threshold $T$, we return the certification-network output, otherwise we pass the sample through the core-network.

**Certification** Using convex relaxation-based certification methods requires sound over-approximations of all layers. To derive an approximation of the entropy, we first recast Equation 4, including softmax and introduce the log-sum-exp trick to improve numerical stability, as

$$H(p(\boldsymbol{y}_b)) = c + \log(\sum_i e^{y_{b,i}-c}) - \sum_j y_{b,j} \exp(y_{b,j} - c - \log(\sum_i e^{y_{b,i}-c})). \tag{5}$$

We provide a proof of this identity in Appendix H.6. We can now construct an entropy transformer from element-wise transformations corresponding to individual operations in Equation 5. In this work we use intervals and zonotopes (Appendix H), however, one can explore other approximations.

**Joint Training** Using vanilla provable training for the certification-network leads to a significant dependence of the entropy on the difficulty of a sample perturbation. This causes a wide range of possible entropies over the admissible perturbations, reducing the robustness of a thresholding based selection. To address this, we would like to decrease the width of the entropy range, while ideally decreasing the entropy for certifiable samples and increasing it for non-certifiable samples. We do this by introducing an entropy loss term $\mathcal{L}_H(\boldsymbol{y}_b, y_s(\boldsymbol{x})) = \text{sign}(y_s(\boldsymbol{x})) \cdot H(\text{softmax}(\boldsymbol{y}_b))$ using the convention $\text{sign}(0) = -1$, and the weighting factor $\lambda$:

$$\mathcal{L}_{joint}(h_{\theta_b}^b, \boldsymbol{x}, y) = (1 - \lambda) \cdot \mathcal{L}_{CE}(\boldsymbol{y}_b, y) + \lambda \cdot \mathcal{L}_H(\boldsymbol{y}_b, y_s(h_{\theta_b}^b, \boldsymbol{x})). \tag{6}$$

We replace the cross entropy loss $\mathcal{L}_{CE}$ with the joint loss $\mathcal{L}_{joint}$ enabling adversarial and provable training against perturbations targeting both classification and entropy, improving selection robustness. We train the certification-network with this joint loss, completely independent from the core network. The selection target $y_s(h_{\theta_b}^b, \boldsymbol{x})$ for the natural, adversarial, and robust losses is, unlike in Equation 3, computed based on the natural, adversarial, and certifiable correctness of the certification-network, respectively, and not always on the certifiable correctness. Using this loss with networks of low accuracy has the disadvantage that the entropy loss encourages a more ambiguous output distribution for samples that are not classified (provably) correctly. Therefore, we perform pretraining with $\lambda = 0$.

### 4.2 END-TO-END CERTIFICATION

After the network is trained, we need to prove that the classification by the compositional network

$$y = \arg\max_j \left[ g_{\theta_s}(\boldsymbol{x}') \cdot h_{\theta_b}^b(\boldsymbol{x}') + (1 - g_{\theta_s}(\boldsymbol{x}')) \cdot h_{\theta_t}^t(\boldsymbol{x}') \right]_j, \quad \forall \boldsymbol{x}' \in B_\epsilon^\infty(\boldsymbol{x}) \tag{7}$$

is correct and robust (as defined by Equation 1). Using one of the certification methods introduced in Section 3 to instantiate the certification function $\text{cert}(\mathbb{X}, f, y)$, the proof of robustness can be broken down into the following steps:

1. Evaluate the selection mechanism $g_{\theta_s}$ on the unperturbed sample $\boldsymbol{x}$ resulting in $g_{\theta_s}(\boldsymbol{x})$.

2. Certify robustness of the decision of the selection mechanism $\mathrm{cert}(B_\epsilon^\infty(\boldsymbol{x}), g_{\theta_s}, g_{\theta_s}(\boldsymbol{x}))$.

3. If this certification was successful, certify the network selected for the unperturbed sample, otherwise certify both the core- and the certification-network $\mathrm{cert}(B_\epsilon^\infty(\boldsymbol{x}), h_{\theta_{\{b,t\}}}^{\{b,t\}}, y)$.

The deep core-network is typically not trained for certifiability and certification is often computationally infeasible. Therefore, we assume that certification of the core-network always fails. Consequently, only samples with a positive natural selection decision can be certified, making certification independent of the core-network. The results of the first and the second step can be used to determine which network will classify the unperturbed sample and which networks could classify a perturbed sample. This information can be used to compute the natural and adversarial accuracy of the combined network without evaluating them jointly.

## 5 EXPERIMENTAL EVALUATION

In this section, we demonstrate the effectiveness of ACE by showing that existing certified defenses cannot achieve the high natural accuracies at non-trivial provable accuracies that we obtain.

**Models and Datasets** We evaluate ACE on 3 different certification-network architectures similar to the models used in Gowal et al. (2018) and Balunović & Vechev (2020), on CIFAR-10, ImageNet200, and TinyImageNet with $\ell_\infty$ perturbations between 1/255 and 8/255, reporting Top-1 accuracies. TinyImageNet is a selection of 200 classes from ImageNet with samples cropped to meaningful regions of the image and downscaled to $64 \times 64$. ImageNet200 is the full sized ImageNet restricted to the same 200 classes but always center cropped for evaluation. We denote as Conv2, Conv3 and Conv5 feed-forward networks with 2, 3, and 5 convolutional layers, respectively. Conv3 corresponds to the largest network from Balunović & Vechev (2020), and Conv5 corresponds to the largest network from Zhang et al. (2020), DM-Large. More details can be found in Appendix A, Table 3. We use an adversarially trained EfficientNet-B0 (Tan & Le, 2019) with ImageNet pretraining as a core-network, using adversarial instead of natural training, as we believe empirical robustness to also be relevant in domains where deterministic guarantees are desired.

**Training and Certification** We perform all experiments, with the exception of reference network training, on a single GeForce RTX 2080 Ti GPU and implement training and certification in PyTorch (Paszke et al., 2019). We train selection- and certification-networks using IBP (Gowal et al., 2018), CROWN-IBP (Zhang et al., 2020) and COLT (Balunović & Vechev, 2020). The hyperparameters can be found in Appendix B. We use adversarial pretraining for COLT trained models. For Entropy Selection we set the joint loss factor to $\lambda = 0.5$. We only use the relatively fast, convex relaxation-based certification methods IBP (Gowal et al., 2018), CROWN-IBP (Zhang et al., 2020), and DeepZ (Singh et al., 2018) for IBP and COLT trained networks respectively, unless specified otherwise. We use 40 step PGD to evaluate adversarial accuracy, using the strategy described in Appendix E to avoid gradient masking effects (Papernot et al., 2017).

**Comparison with Existing Architectures** We show that ACE can, in contrast to state-of-the-art provable training methods, achieve both high natural and non-trivial provable accuracies, by enabling an efficient trade-off between natural and certifiable accuracy instead of maximizing provable robustness at any cost. As direct comparison with prior work is difficult, we show that using our best effort we were not able to achieve comparable combinations of natural and certifiable accuracy using the state-of-the-art COLT and CROWN-IBP provable training methods.

COLT is computationally expensive, restricting it to relatively small models. We train the biggest model evaluated by Balunović & Vechev (2020), Conv3, on CIFAR-10 at $\epsilon_\infty = 2/255$ using COLT with a varying natural loss component and use DeepZ for certification. In Figure 2 we show certified vs natural accuracy for each of these models (yellow points). To compare with our approach, we train an ACE model (teal squares) using one of these networks (teal triangle) as certification-network. We observe that the ACE model obtains higher certified accuracies at all natural accuracies, still yielding a certified accuracy of 36.8% at the highest natural accuracy (85.1%) obtained using an individual, naturally trained Conv3 network. Using the Conv3 model trained by Balunović & Vechev (2020)

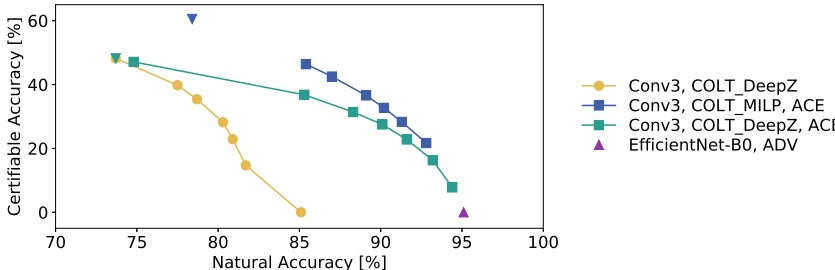

Figure 2: Natural and certified accuracy of different COLT trained models on CIFAR-10 with $\epsilon_\infty = 2/255$. We compare individual Conv3 networks (yellow dots), trained with COLT and varying natural loss components, with different ACE models (squares) based on an EfficientNet-B0 core-network (purple) and different certification-networks (triangles): Conv3 with DeepZ certification (teal) and Conv3 with MILP certification (blue). Further up and to the right is better. The horizontal distance between the yellow and teal line is the increase in natural accuracy due to using ACE instead of changing the natural loss component int COLT training.

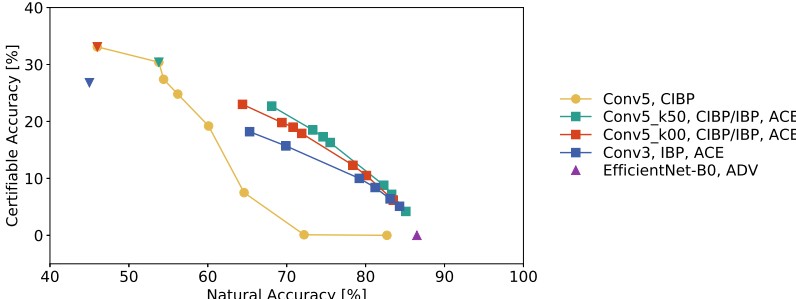

Figure 3: Natural and certified accuracy on CIFAR-10 with $\epsilon_\infty = 8/255$. We compare individual Conv5 networks (yellow dots), trained with CROWN-IBP and varying natural loss components, with different ACE models (squares) based on an EfficientNet-B0 core-network (purple) and different certification-networks (triangles): CROWN-IBP trained Conv5 from Zhang et al. (2020) (red), CROWN-IBP trained Conv5 with $\kappa_{end} = 0.5$ (teal) and IBP trained Conv3 (blue). All selection-networks are IBP trained.

(blue triangle) in combination with MILP certification as certification-network, we obtain an even stronger ACE model (blue squares). We compare to CROWN-IBP trained reference networks in Appendix C, yielding the same conclusion.

CROWN-IBP can be applied to larger models, includes an inherent robustness-accuracy trade-off parameter $\kappa$, weighting the natural and robust loss components, and outperforms COLT on CIFAR-10 at $\epsilon_\infty = 8/255$, making it the perfect benchmark for these larger perturbations. Using the original implementation and the largest model Zhang et al. (2020) evaluate on CIFAR-10, Conv5, we vary $\kappa_{end}$ to obtain several models. We show certified vs natural accuracy for each of these CROWN-IBP models (yellow points) in Figure 3. Using the Conv5 network published by Zhang et al. (2020) with $\kappa_{end} = 0.0$ (red triangle) and one we trained with $\kappa_{end} = 0.5$ (teal triangle), we train ACE models (red and teal squares) using IBP for the selection network training, which both outperform the individual Conv5 networks over a wide range of natural accuracies. Even when using a much weaker certification-network, such as an IBP trained Conv3 (blue triangle), we obtain an ACE model (blue squares) yielding more attractive trade-offs at high natural accuracies.

These results show that across different provable training and certification methods, network architectures and perturbation sizes, ACE produces much more favorable robustness-accuracy trade-offs than varying hyperparameters of existing certified defenses. ACE models can always use a certification-network trained at the efficiency sweetspot of the employed provable training method, allowing any improvements in certified defenses to be utilized, while allowing for flexibility in the trade-off between accuracy and robustness. As ACE is truly orthogonal to all of these methods, it should be seen as a complement to and not a replacement for provable training methods.

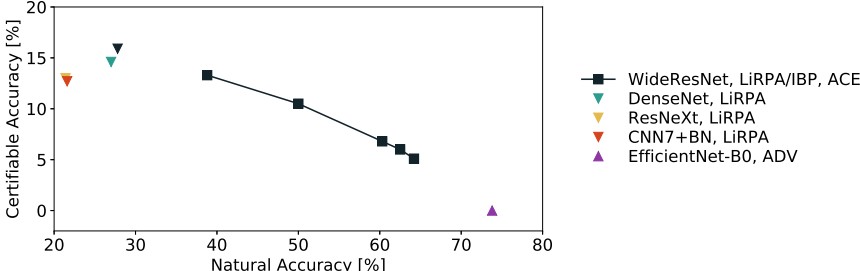

Figure 4: Natural and certified accuracy on TinyImageNet with $\epsilon_\infty = 1/255$. We compare the LiRPA trained networks from Xu et al. (2020) WideResNet (black triangle), DenseNet (teal), ResNeXt (yellow), and CNN7+BN (red) with an ACE model (black squares) using the same LiRPA trained WRN as certification-network and as feature extractor for the otherwise IBP trained selection-network. The ACE model uses an EfficientNet-B0 core-network (purple).

Table 1: Natural, adversarial and certifiable accuracy for various ACE models. The training methods COLT, LiRPA with loss fusion, IBP and CROWN-IBP (C-IBP) used for the training of certification- (CERT) and selection-networks (SELECT) are indicated separately.

| Dataset | $\epsilon_\infty$ | Selection Method | ACE Training | | Certification Network | Top-1 Model Accuracy [%] | | |
| | | | CERT | SELECT | | Natural | Adversarial | Certified |
|---|---|---|---|---|---|---|---|---|
| CIFAR-10 | $\frac{2}{255}$ | SelectionNet | COLT | COLT | Conv3 | 90.1 | 78.4 | 27.5 |
| | $\frac{8}{255}$ | SelectionNet | C-IBP | IBP | Conv5 | 80.1 | 48.8 | 10.5 |
| ImageNet200 | $\frac{1}{255}$ | SelectionNet | COLT | COLT | Conv3 | 70.0 | 60.5 | 3.1 |
| TinyImageNet | $\frac{1}{255}$ | SelectionNet | LiRPA | IBP$^\dagger$ | WRN | 50.0 | 35.9 | 10.5 |

$^\dagger$ LiRPA trained WideResNet from Xu et al. (2020) used as feature extractor for the selection-network.

The compositional structure of ACE has the additional advantage of permitting every component network to work at a different resolution. For tasks where high resolution images are available, the core network can process full-scale images, while down-scaled versions can be passed through the selection and certification network. For ImageNet200, we use full-scale images in the core-network to obtain a high natural (70.0%) accuracies, while the certification- and selection-network yields a non-trivial certifiable accuracy (3.1%) at $\epsilon_\infty = 1/255$ using samples scaled down to $64 \times 64$. Here, the certifiable accuracy is limited by the lack of a strong certifiably robust network to use as a certification-network.

For TinyImageNet no full size images are available, reducing the advantage of an ACE model. However using the LiRPA trained WideResNet (black triangle in Figure 4) from Xu et al. (2020) as certification-network and feature extractor for the otherwise IBP trained selection-network, we train an ACE model (black squares) showing a very similar trade-off characteristic as the CIFAR-10 models, demonstrating that ACE scales to larger tasks.

In Table 1 we present results on the CIFAR-10 and TinyImageNet datasets for selected models. An extended table showing more results can be found in Appendix G.

**Selection** Recall that the key to our compositional architecture is a provably robust selection mechanism that can differentiate samples based on their certifiability by the certification-network. We try to certify the natural selection decision made by a Conv3 selection-network on CIFAR-10 at $\epsilon_\infty = 2/255$ and split samples into three groups: samples selected for all admissible perturbations (provably selected), samples not selected for any admissible perturbation (provably non-selected), and the remainder for which we cannot prove either decision

Table 2: Certifiable accuracy of the certification-network depending on selection decision using a COLT trained Conv3 selection- and certification-network for CIFAR-10 at $\epsilon_\infty = 2/255$.

| | Certifiable Accuracy [%] |
|---|---|
| provably selected | 72.9 |
| non-provably selected | 52.3 |
| not selected | 28.4 |
| full test set | 47.9 |

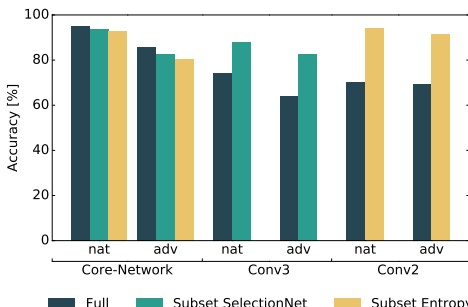
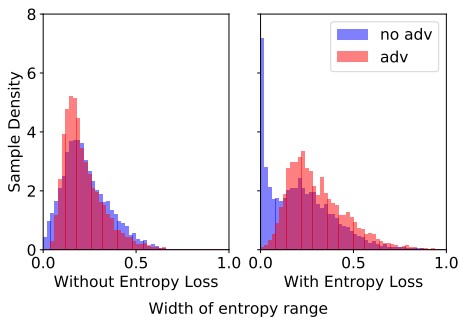

Figure 5: Natural and adversarial accuracy for two ACE networks: one with SelectionNet + Conv3 and one with entropy selection + Conv2. We evaluate classification networks on the full test set and its subsets selected by the two selection mechanisms.

Figure 6: Width of the entropy range over admissible perturbations. Samples are denoted as adversarial if we can successfully attack the certification-network and non-adversarial otherwise.

(non-provably selected). In Table 2 we show that for the provably selected samples the certifiable accuracy (72.9%) is much higher than on the full test set (47.9%), while it is much lower for the provably non-selected samples (28.4%). This shows that the selection-network successfully separates samples based on certification difficulty.

**Evaluating Core- and Certification-Networks**   Next, we train two ACE networks for CIFAR-10 with $\epsilon_\infty = 2/255$ using COLT: one with a selection-network and Conv3 certification-network, and another with entropy selection and a Conv2 certification-network. Both use EfficientNet-B0 as a core-network. We evaluate the natural and adversarial accuracy of both the core- and certification-network on the full test set and its subsets selected by the selection mechanism. The results are shown in Figure 5. We observe that the accuracy of both certification-networks is significantly higher on their respective selected datasets, while the accuracy of the core-networks decreases. This indicates that the selection mechanism can successfully separate samples by classification difficulty and assign easier samples to the certification- and more difficult samples to the core-network.

**Effectiveness of the Entropy Loss**   Recall that we introduced the entropy loss to make Entropy Selection more robust, by decreasing the sensitivity to different perturbations. To assess its effectiveness, we train two Conv2 certification-networks using COLT for CIFAR-10 with $\epsilon_\infty = 2/255$, with and without entropy loss. Note that using entropy loss corresponds to $\lambda = 0.5$ and not using it corresponds to $\lambda = 0.0$ in Equation 6. We split the test set into two groups based on whether an adversarial attack on the certification-network is successful (adversarial) or not (non-adversarial). For each sample, we compute the difference between the largest and the smallest entropy that can be obtained by perturbing it. Figure 6 shows histograms of these differences (or widths) for both the adversarial and non-adversarial group. Clearly, the non-adversarial samples lead to much narrower entropy ranges if an entropy loss was used, while there is no significant difference if no entropy loss is used. This demonstrates that the entropy loss successfully increased the robustness of the entropy selection mechanism for non-adversarial samples (as adversarial ones are not certifiable anyway).

## 6   CONCLUSION

We proposed a new architecture that boosts the certifiable robustness of any state-of-the-art network, while retaining high accuracy and without requiring retraining. The key idea is to combine this network with a provably trained certification-network and a certifiable selection mechanism, which adaptively decides at inference-time which of the two networks to use. We presented two such selection mechanisms with corresponding training and certification methods. Our experiments show that using this method, one can achieve both high natural accuracies and non-trivial certifiable robustness, beyond the reach of state-of-the-art certified defenses. Our architecture is also fully orthogonal to certified defenses, allowing any advances in this field to be carried over directly.

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

## A    NETWORK ARCHITECTURES

In Table 3 we list the detailed architectures for CIFAR-10 and ImageNet200, respectively, used in the experiments described in Section 5. The Conv3 architecture for CIFAR-10 is identical to the largest network from Balunović & Vechev (2020).

Table 3: Network architectures of the certification- and selection-networks used for CIFAR-10 and ImageNet200 (Conv3 (IN)). All layers are followed by a ReLU activation. The last fully connected layer is omitted. For Conv3 (IN) a global average pooling layer precedes this last linear layer. "CONV c h×w/s" corresponds to a 2D convolution with c output channels, a h×w kernel size, and a stride of s in both dimensions.

| Conv2 | Conv3 | Conv5 | Conv3 (IN) |
|---|---|---|---|
| CONV 32 4×4/2 | CONV 32 3×3/1 | CONV 64 3×3/1 | CONV 32 5×5/2 |
| CONV 32 4×4/2 | CONV 32 4×4/2 | CONV 64 3×3/1 | CONV 64 5×5/2 |
| FC 200 | CONV 128 4×4/2 | CONV 128 3×3/2 | CONV 128 5×5/2 |
| | FC 250 | CONV 128 3×3/1 | |
| | | CONV 128 3×3/1 | |
| | | FC 512 | |

## B    TRAINING HYPERPARAMETERS

**CIFAR-10 Training**    For CIFAR-10 IBP training is conducted for 200 epochs, annealing $\epsilon$ and $\kappa$ over the first 100 epochs, with an initial learning rate of 1e-3, reducing by half every 10 epochs after annealing is completed. We choose $\kappa_{end} = 0.5$ for all models. COLT training is conducted for 40 epochs per stage with an initial learning rate of 1e-3, decreasing by a factor of 0.75 between stages and by a factor of 0.5 every 5 epochs after an initial loss mixing period of 5 epochs.

**ImageNet Training**    For IBP training on ImageNet200 and TinyImageNet we train for 50 epochs using a batch size of 100, an initial learning rate of 1e-3, reducing it by half every 5 epochs after annealing both $\kappa$ and $\epsilon$ for 10 epochs. We choose $\kappa_{end} = 0.5$ for all models. For COLT training on ImageNet200 we use feature extraction and freeze all weights up to the last linear layer. We train for 20 epochs per stage, using an initial learning rate of 1e-3, reduced by a factor of 0.75 between stages and 0.5 every 3 epochs after an initial loss mixing period of 5 epochs.

## C    ADDITIONAL EXPERIMENTS COMPARING WITH CROWN-IBP

In this section we present additional experiments comparing ACE models with CROWN-IBP.

CROWN-IBP trained networks can not match the performance of COLT trained and MILP certified networks on CIFAR-10 at $\epsilon_\infty = 2/255$ (Balunović & Vechev, 2020; Zhang et al., 2020). However, as their certification is notably cheaper, we still compare their approach in isolation to an ACE compositional using the largest model Zhang et al. (2020) evaluate on CIFAR-10, Conv5. We follow their instructions for multi GPU training and obtain several models by varying the parameter $\kappa_{end}$ (yellow dots in Figure 7), obtaining very similar results for the settings they report ($\kappa_{end} \in \{0, 0.5\}$). Note that every setpoint requires about 4 GPU days, while training an ACE model on top of an available certification-network only takes few hours and new setpoints can be evaluated in minutes. In Figure 7 we show certifiable vs natural accuracy for each of these CROWN-IBP models (yellow points). To compare it with our method, we chose the CROWN-IBP trained Conv5 with $\kappa_{end} = 0.5$ as certification-network (teal triangle) and use it to train our ACE models with various thresholds $T$, shown as teal squares in Figure 7. Clearly, our models achieve much better robustness-accuracy trade-offs, especially in regions of high natural accuracy. We do not compare the CROWN-IBP results to those obtained using MILP in the abstract, as the latter requires a much more expensive certification approach. However, even when using a much weaker certification network, such as an IBP trained Conv3 (black triangle), we can still train ACE models (black squares) obtaining more attractive trade-offs at high natural accuracies.

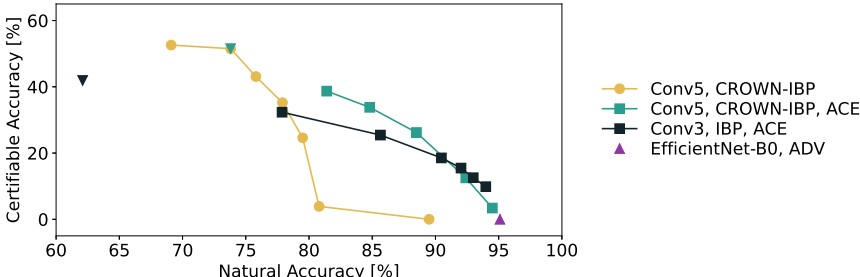

Figure 7: Natural and certified accuracy on CIFAR-10 with $\epsilon_\infty = 2/255$. We compare individual Conv5 networks (yellow dots), trained with CROWN-IBP and varying natural loss components, with different ACE models (squares) based on an EfficientNet-B0 core-network (purple) and different certification-networks (triangles): CROWN-IBP trained Conv5 (teal) and IBP trained Conv3 (black). Further up and to the right is better.

## D  ADDITIONAL EXPERIMENTS ON COLT TRAINED ACE MODELS

In this section we describe additional experiments using COLT trained Conv3 networks.

We use the Conv3 network published by Balunović & Vechev (2020) and certify it using DeepZ (Singh et al., 2018) and MILP (Tjeng et al., 2017). To unlock the full potential of this MILP certified certificaiton network, we would ideally want to train our selection network with MILP based labels. However, certifying the whole or a significant portion of the training set using MILP is infeasible. Therefore, we have to use surrogate labels when training the selection-network. We evaluate three approaches to training a selection network and present the results in Figure 8:

- Transfer the selection network of a different ACE model based on a certification-network with higher zonotope certified accuracy (teal)
- Compute selection labels using zonotope certification (brown)
- Compute selection labels using adversarial correctness (yellow)

We observe that using the last approach consistently, over a range of selection rates, performs worse than both other methods, suggesting that the selection network can learn features that make a sample hard to certify while not necessarily leading to successful adversarial attacks. Interestingly, the selection network transferred from a different Conv3 network (with a higher zonotope certified accuracy), does not only help to improve the performance of the ACE model at high natural accuracies when certified using MILP, but also when using DeepZ. This suggests that the properties making a sample difficult to certify show at least some level of stability over different networks.

## E  ADVERSARIAL ACCURACY COMPUTATION

The adversarial accuracies listed in Table 1 are intended to be purely informative and not to be considered as a strong indicator of the true robust accuracy. Nevertheless, we consider the potential problem of gradient masking (Papernot et al., 2017) caused by the compositional structure and developed the following three approaches to calculate adversarial accuracy:

- We compute separate adversarial samples attacking the core-, selection- and certification-network. If we find any perturbation leading to the selection of a classification network, we consider it reachable and evaluate the corresponding adversarial example. Only if no attack on a reachable network is successful, we consider the adversarial attack to have failed. Note, that a successful adversarial example to either classification network would not necessarily be classified by this network. Therefore, this approach does not provide a true upper bound to the robust accuracy.
- We attack the classification networks as described above, but use certifiable instead of empirical reachability, that is consider a network reachable unless we can proof that it can not be reached. This approach provides an even more conservative adversarial accuracy.

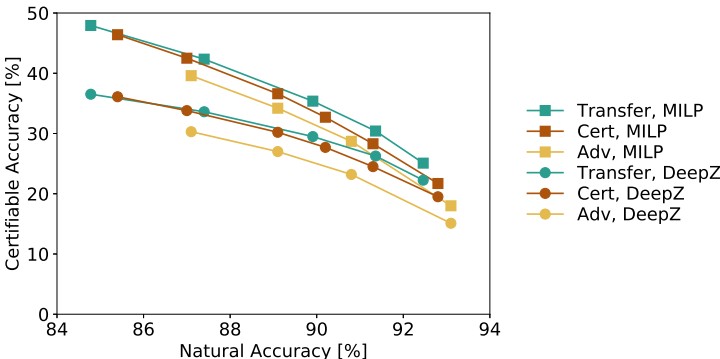

Figure 8: Natural and certified accuracy on CIFAR-10 at $\epsilon_\infty = 2/255$. ACE networks using a COLT trained Conv3 network directly from Balunović & Vechev (2020) as certification network and three different selection networks, certified using DeepZ (dots) or MILP (squares).Selection networks transferred from a different Conv3 network (teal), trained on certifiable correctness (brown) and on adversarial correctness (yellow).

- We compute two adversarial attacks. One aimed at the core- and the other at the certification-network. To this end, we combine the classification networks loss term with a loss component from the selection network, designed to perturb the sample such that the currently targeted network gets selected. We only consider an attack successful, if an adversarial example gets classified incorrectly by the compositional network. To reduce the gradient obfuscation problem in this setting, we weight the selection network loss term based on whether we already select the currently targeted network.

While the gap between the first two approaches is usually very small at less than 1%, the third approach sometimes yields notably higher adversarial accuracies. Therefore, we decided to report the most conservative numbers, obtained using the second approach. We use PGD with 40 steps, a step size of $0.035\,\epsilon$ and one restart with a random initialization of the perturbation.

## F  NATURAL AND ADVERSARIAL CORE NETWORKS

In this section, we compare ACE models obtained using naturally and adversarially trained core-networks.

In both cases we use an EfficientNet-B0 with ImageNet pretraining. On CIFAR-10 at $\epsilon_\infty = 2/255$, natural training leads to a natural accuracy of 97.4% but only 6.7% of adversarial accuracy, while adversarial training yields 95.1% and 85.6%, respectively. In Figure 9 we show certified over natural accuracy and compare individual Conv3 networks (yellow dots), trained with COLT and varying natural loss components, with ACE models (squares) using the same Conv3 certification-network (triangles), but different core-networks. The ACE model with a naturally trained core-network is shown in blue and that with an adversarially trained core-network in teal. At very high natural accuracies the relative drop in natural accuracy from core-network to ACE model is significantly higher when a naturally trained core-network is used. This permits a much higher selection rate, leading to a significant increase in certified accuracy for any given natural accuracy. Despite these improvements, we maintain that using an adversarially trained core network is more representative of potential real world usage, where empirical robustness guarantees are also considered.

## G  RESULTS ON SELECTED SETPOINTS

In Table 4 we present selected setpoints of ACE models trained on CIFAR-10, TinyImageNet and ImageNet200, demonstrating that ACE can be applied to a range of models, datasets and provable training methods.

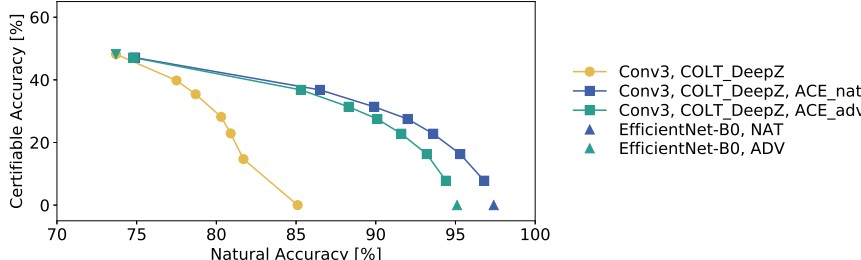

Figure 9: Natural and certified accuracy of different COLT trained models on CIFAR-10 with $\epsilon_\infty = 2/255$. We compare individual Conv3 networks (yellow dots), trained with COLT and varying natural loss components, with ACE models (squares) using the same Conv3 certification-network (triangles), but different core-networks: a naturally trained EfficientNet-B0 (blue) and an adversarially trained EfficientNet-B0 (teal).

Table 4: Natural, adversarial and certifiable accuracy for various ACE models. The training methods COLT, LiRPA with loss fusion, IBP and CROWN-IBP (C-IBP) used for the training of certification-(CERT) and selection-networks (SELECT) are indicated separately.

| Dataset | $\epsilon_\infty$ | Selection Method | ACE Training CERT | SELECT | Certification Network | Top-1 Model Accuracy [%] Natural | Adversarial | Certified |
|---|---|---|---|---|---|---|---|---|
| CIFAR-10 | $\frac{2}{255}$ | SelectionNet | COLT | COLT | Conv2 | 90.4 | 79.0 | 20.5 |
| | | | COLT | COLT | Conv3 | 90.1 | 78.4 | **27.5** |
| | | | IBP | IBP | Conv3 | **90.5** | **80.5** | 18.5 |
| | | | C-IBP | C-IBP | Conv5 | 88.6 | 77.6 | 25.6 |
| | | Entropy | COLT | - | Conv2 | **93.4** | **85.1** | **19.4**[†] |
| | | | IBP | - | Conv2 | 90.2 | 71.7 | 7.2 |
| | $\frac{8}{255}$ | SelectionNet | COLT | COLT | Conv3 | 77.1 | 46.7 | 7.4 |
| | | | IBP | IBP | Conv3 | **83.1** | **49.9** | 6.4 |
| | | | C-IBP | IBP | Conv5 | 80.1 | 48.8 | **10.5** |
| ImageNet200 | $\frac{1}{255}$ | SelectionNet | COLT | COLT | Conv3 | **70.0** | **60.5** | 3.1 |
| | | | IBP | IBP | Conv3 | 68.3 | 57.4 | **3.8** |
| TinyImageNet | $\frac{1}{255}$ | SelectionNet | LiRPA | IBP[‡] | WRN | **50.0** | **35.9** | 10.5 |

[†] Evaluated using MILP certification (Tjeng et al., 2017) on the first 1000 samples of the test set.
[‡] LiRPA trained WideResNet from Xu et al. (2020) used as feature extractor for the selection-network.

## H  ZONOTOPE TRANSFORMERS

For both training and certification with convex relaxations it is essential to have precise, also called tight, transformers to reduce the accumulation of errors. Singh et al. (2018) provide such transformers for the ReLU, tanh and sigmoid function. In this section a general approach to constructing transformers for the zonotope domain and $\mathbb{C}^1$ continuous, concave or convex functions will be described and transformers for the exponential and logarithm function, and the product of two zonotopes introduced. These can then be combined into a transformer for the entropy function.

### H.1  ZONOTOPE DOMAIN

The zonotope domain Ghorbal et al. (2009) is a classic numeric abstract domain, shown to be suitable as convex relaxation for analyzing neural networks Gehr et al. (2018), as it is exact for affine transformations and efficient abstract transformers for the ReLU, sigmoid and tanh function exist Singh et al. (2018). A zonotope $\mathcal{Z} \subseteq \mathbb{R}^n$ approximation of a n-dimensional variable $x \in \mathbb{R}^n$ is described by an affine form $\hat{x}_j$ for every dimension $x_j$ as

$$\hat{x}_j = a_{j,0} + \sum_{i=1}^{p} a_{j,i} \cdot \epsilon_i, \qquad a_{j,i} \in \mathbb{R}, \ \epsilon_i \in [-1,1] \tag{8}$$

with the center coefficient $a_{j,0}$, error coefficients $a_{j,i}$ and shared error terms $\epsilon_i$. These shared error terms allow the representation of implicit dependencies between dimensions and make the zonotope domain strictly more powerful then the interval domain (equivalence is given for a diagonal error matrix). Further, the matrix notation

$$\hat{\boldsymbol{x}} = \boldsymbol{a}_0 + \boldsymbol{A}\epsilon, \qquad \boldsymbol{a}_0 \in \mathbb{R}^{n \times 1}, \boldsymbol{A} \in \mathbb{R}^{n \times p}, \epsilon \in [-1,1]^{p \times 1} \tag{9}$$

makes affine transformations of the form $\boldsymbol{y} = \boldsymbol{B}\boldsymbol{x} + \boldsymbol{c}$ very simple to apply:

$$\hat{\boldsymbol{y}} = \underbrace{\boldsymbol{B}\,\boldsymbol{a}_{0,in} + \boldsymbol{c}}_{\boldsymbol{a}_{0,out}} + \underbrace{(\boldsymbol{B}\boldsymbol{A}_{in})}_{\boldsymbol{A}_{out}}\epsilon = \boldsymbol{a}_{0,in} + \boldsymbol{A}_{out}\epsilon \tag{10}$$

## H.2 GENERAL TRANSFORMER CONSTRUCTION

Sound neuron-wise transformers for the zonotope domain can be visualized in the input-output-plane of the to be approximated function as parallelograms with vertical left and right edges, fully enclosing the function on the input interval. They can be described as

$$\hat{y}_j = \lambda_j \hat{x}_j + \xi_j + \mu_j \epsilon_{p+1} \tag{11}$$

for the $j^{\text{th}}$ dimension of the input zonotope $\hat{\boldsymbol{x}}$ with $p$ error terms and the neuron-/dimension-wise parameters slope $\lambda_j$, offset $\xi_j$ and looseness $\mu_j$

The height of the parallelogram $2\mu_j$ corresponds to the magnitude of the new error term. As the width is only dependent on the range of the input, the parallelogram's area is only dependent on this height. A transformer is strictly more precise than another, if the parallelogram representation of the former is fully enclosed in the one of the latter. While a smaller area generally corresponds to a smaller loss of precision, no guarantees can be given.

**Viability** shall be defined as the absence of strictly more precise transformers of the same form. The looseness $\mu$ of viable transformers is uniquely determined by the slope $\lambda$, as the offset $\xi$ and $\mu$ can be chosen so that both the upper and lower edges touch the function plot in at least one point. All transformers with different offsets and looseness but the same slope will enclose this viable transformer and are therefore strictly less precise.

If one of these contact point lies within the input interval $x \in [x_{lb}, x_{ub}]$, the slope is by definition a subgradients to the function at this point. If one contact points lie on the borders of the input interval, the slope can be reduced/increased in the direction of the local one-sided gradient until either it is the tangent to that point or an additional contact point is made. The new slope is in both cases by definition a subgradient to the function on the input domain at both contact points. This new transformer is also strictly more precise than the original one. It follows, that all viable slopes are subgradients to the function at one point in the interval.

**Convexity** can be assumed without loss of generality for convex and concave functions as the latter can be negated to ensure convexity. For convex, $C^1$ continuous functions all tangents to the curve of the function yield viable transformers, consequently they can be parametrized by the x-position $x_{lb} \leq t \leq x_{ub}$ of the contact point. Using the mean value theorem and convexity it follows that there will be a point $t_{crit}$ where the upper edge of the parallelogram will connect the lower and upper endpoints of the graph. For $t < t_{crit}$ it will make contact on the upper endpoint and for $t > t_{crit}$ on the lower endpoint. This allows to describe the parameters $\lambda$, $\xi$ and $\mu$ of a zonotope transformer for an element-wise function $f(x) : \mathbb{R} \to \mathbb{R}$ on the interval $[x_{lb}, x_{ub}]$ as

$$\lambda = f'(t) \tag{12}$$

$$\xi = \frac{1}{2}\left( f(t) - \lambda t + \begin{cases} f(x_{lb}) - \lambda x_{lb}, & \text{if } t \geq t_{crit} \\ f(x_{ub}) - \lambda x_{ub}, & \text{if } t < t_{crit} \end{cases} \right) \tag{13}$$

$$\mu = \frac{1}{2}\left( \lambda t - f(t) + \begin{cases} f(x_{lb}) - \lambda x_{lb}, & \text{if } t \geq t_{crit} \\ f(x_{ub}) - \lambda x_{ub}, & \text{if } t < t_{crit} \end{cases} \right) \tag{14}$$

$$\nabla_x f(x)|_{x=t_{crit}} = \frac{f(x_{ub}) - f(x_{lb})}{x_{ub} - x_{lb}} \tag{15}$$

**Minimum Area Transformer** – A minimum area transformer can now be derived by minimizing the looseness $\mu$ for $x_{lb} \leq t \leq t_{crit}$ and $t_{crit} \leq t \leq x_{ub}$. This yields the constrained optimization problems:

$$\min_t \frac{f'(t)(t - x_{ub}) - f(t) + f(x_{ub})}{2}, \qquad s.t. \quad t \geq x_{lb}, \quad t \leq t_{crit} \tag{16}$$

$$\min_t \frac{f'(t)(t - x_{lb}) - f(t) + f(x_{lb})}{2}, \qquad s.t. \quad t \geq t_{crit}, \quad t \leq x_{ub} \tag{17}$$

These can be solved using the method of Lagrange multipliers. Equation 16 yields the Lagrangian function:

$$\mathcal{L}(t, \gamma) = \frac{1}{2}(f'(t)(t - x_{ub}) - f(t) + f(x_{ub})) - \gamma_1(t - x_{lb}) + \gamma_2(t - t_{crit}) \tag{18}$$

$$\nabla_t \mathcal{L}(t, \gamma) = \frac{1}{2}(f''(t)(t - x_{ub}) + \underline{f'(t)(1 - 1)}) - \gamma_1 + \gamma_2 \overset{!}{=} 0 \tag{19}$$

$$\nabla_{\gamma_1} \mathcal{L}(t, \gamma) = t - x_{lb} \overset{!}{=} 0 \tag{20}$$

$$\nabla_{\gamma_2} \mathcal{L}(t, \gamma) = t - t_{crit} \overset{!}{=} 0 \tag{21}$$

As $x_{lb} < t_{crit} < x_{ub}$, at most one of the two constraints can be active at any time. This yields three cases:

**Case 1:** neither constraint is active, $\gamma_1 = \gamma_2 = 0$

$$\nabla_t \mathcal{L}(t, \gamma) = \underbrace{f''(t)}_{\substack{\geq 0 \text{ convex} \\ > 0 \text{ strictly convex}}} (t - x_{ub}) = 0$$

$$t_1 = x_{ub} \quad \notsubset \quad t \leq t_{crit}$$

$$f''(t_2) = 0 \quad \Rightarrow \text{ saddlepoint}$$

**Case 2:** $t = x_{lb}, \gamma_1 \neq 0, \gamma_2 = 0$

$$\gamma_1 = \frac{1}{2}(f''(x_{lb})(x_{lb} - x_{ub})$$

$$t_3 = x_{lb} \quad \notsubset \quad \nabla_t \mu(t)|_{t=x_{lb}} = \underbrace{f''(x_{lb})}_{\geq 0} \underbrace{(x_{lb} - x_{ub})}_{\leq 0} \leq 0 \Rightarrow \text{boundary maximum}$$

**Case 3:** $t = t_{crit}, \gamma_1 = 0, \gamma_2 \neq 0$

$$\gamma_2 = \frac{1}{2}(f''(x_{lb})(x_{lb} - x_{ub})$$

$$t_4 = t_{crit} \quad \nabla_t \mu(t)|_{t=t_{crit}} = \underbrace{f''(t_{crit})}_{\geq 0} \underbrace{(t_{crit} - x_{ub})}_{\leq 0} \leq 0 \Rightarrow \text{boundary minimum}$$

Analogously, equation 17 yields a boundary minimum at $t = t_{crit}$. Consequently $t = t_{crit}$ yields the minimum area transformer for convex functions. $t_{crit}$ can be computed either analytically or numerically by solving equation 15 as the point where the local gradient is equal to the mean gradient over the whole interval. It can be observed that this yields the same slope as the minimum area transformer for the ReLU function Singh et al. (2018) even though this derivation can not be applied there directly due to the ReLU functions $C^1$ discontinuity.

## H.3 Exponential Transformer

The exponential function has the feature that its output is always strictly positive, which is important when used as input to the logarithmic function to compute the entropy. Therefore, a guarantee of positivity for the output zonotope is desirable. A constraint yielding such a guarantee can be obtained by inserting $\hat{x}_j = x_{lb}, \epsilon_{p+1} = -\text{sign}(\mu)$ and $\hat{y}_j \geq 0$ with $\lambda(t) = e^t$ into equation 11:

$$0 \leq \lambda x_{lb} + \frac{1}{2}\big(f(t) - \lambda t + f(x_{ub} - \lambda x_{ub})\big) - \frac{1}{2}\big(\lambda t - f(t) + f(x_{ub} - \lambda x_{ub})\big)$$

$$0 \leq \lambda(x_{lb} - t) + f(t)$$

$$0 \leq e^t(x_{lb} - t + 1)$$

$$t \leq 1 + x_{lb} \equiv t_{crit,2} \tag{22}$$

This constitutes the additional upper limit $t_{crit,2}$ on $t$. Therefore it is sufficient to reevaluate 16 as it will either be inactive in equation 17 if $t_{crit} \leq t_{crit2}$ for the solutions computed previously or the constraints will be insatiable ensuring that 17 will have no solutions. If a strictly positive output is required a small delta can simply be subtracted from the upper limit $t_{crit,2}$. It is easy to see that $t$ is now constrained to $[x_{lb}, \min(x_{ub}, t_{crit,2})]$ and that the minimum area solution will be obtained with $t_{opt} = \min(t_{crit}, t_{crit,2})$. The critical points can be computed explicitly to $t_{crit} = log(\frac{e^{x_{ub}} - e^{x_{lb}}}{x_{ub} - x_{lb}})$ and $t_{crit,2} = x_{lb} + 1$. This can be inserted into equations 11 to 14 to obtain a positive, sound and viable transformer. This transformer is visualized for different choices of $t$ in figure 10.

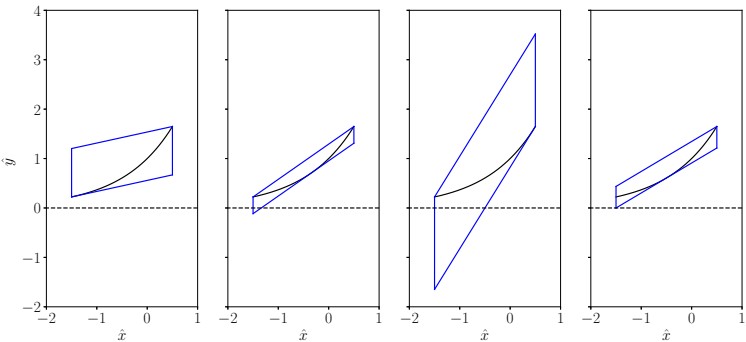

Figure 10: Illustration of the transformer for the exponential function for, from left to right $t = x_{lb}$, minimum area: $t = t_{crit}$, $t = x_{ub}$ and minimum area while strictly positive: $t = t_{crit,2}$.

If there is no point in the input interval where the gradient of the to be approximated function is 0, as is always the case for the exponential function, the box transformer is not a viable zonotope transformer. But the viable transformer with the smallest gradient at $t = x_{lb}$ is strictly more precise than the box transformer (cf. figure 10).

## H.4 LOGARITHMIC TRANSFORMER

The logarithmic transformer can be constructed by plugging $f(t) = -log(t)$ and $f'(t) = \frac{-1}{x}$ into equations 12 to 14 and their results into equation 11. Equation 15 can be solved to $t_{crit} = \frac{x_{lb} - x_{ub}}{ln(x_{lb}) - ln(x_{ub})}$. The resulting transformer is visualized in figure 11. It becomes apparent that the choice of $\lambda$ can have a significant impact on the looseness of the obtained transformer.

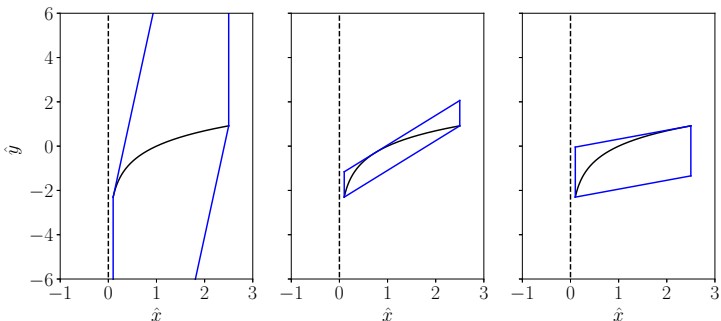

Figure 11: Illustration of the transformer for the logarithmic function for, from left to right $t = x_{lb}$, minimum area: $t = t_{crit}$ and $t = x_{ub}$.

Similar to the exponential transformer, the box transformer is not a viable logarithmic zonotope transformer, but the viable transformer with the smallest gradient at $t = x_{ub}$ is strictly more precise than the box transformer (cf. figure 11).

## H.5 PRODUCT TRANSFORMER

The pointwise or Hadamard product is different from the previously introduced transformers as it involves two zonotopes instead of just one. For this derivation the two one-dimensional zonotopes $\hat{x}$ and $\hat{z}$ with $p$ shared error terms and $k_1$ and $k_2$ individual error terms shall be considered. Typically, error terms will be shared up to a certain index (potentially 0) and all following error terms will be individual to one of the zonotopes. In any case this form can obtained by reordering the error terms and can therefore be assumed without loss of generality.

$$\hat{x} = a_0 + \mathbf{A}_{ind} \begin{bmatrix} \epsilon_1 \\ \vdots \\ \epsilon_p \\ \epsilon_{p+1} \\ \vdots \\ \epsilon_{p+k_1} \end{bmatrix}, \quad \hat{z} = b_0 + \mathbf{B}_{ind} \begin{bmatrix} \epsilon_1 \\ \vdots \\ \epsilon_p \\ \epsilon_{p+k_1+1} \\ \vdots \\ \epsilon_{p+k_1+k_2} \end{bmatrix} \quad a_0, b_0 \in \mathbb{R}, \mathbf{A}_{ind} \in \mathbb{R}^{p+k_1}, \mathbf{B}_{ind} \in \mathbb{R}^{p+k_2}$$

A shared error vector $\epsilon \in$ with $q = p + k_1 + k_2$ error terms can be obtained, by concatenating the individual error terms of the second zonotope $\hat{z}$ to the error vector of the first and padding the error coefficient matrices correspondingly with zeros:

$$\hat{x} = a_0 + \underbrace{\begin{bmatrix} \mathbf{A}_{ind}^\top \\ 0 \\ \vdots \\ 0 \end{bmatrix}^\top}_{\mathbf{A}} \underbrace{\begin{bmatrix} \epsilon_1 \\ \vdots \\ \epsilon_{p+k_1} \\ 0 \\ \vdots \\ 0 \end{bmatrix}}_{\epsilon}, \quad a_0 \in \mathbb{R}, \mathbf{A} \in \mathbb{R}^q, \epsilon \in [-1,1]^q$$

$$\hat{z} = b_0 + \underbrace{\begin{bmatrix} b_1 \\ \vdots \\ b_p \\ 0 \\ \vdots \\ 0 \\ b_{p+1} \\ \vdots \\ b_{p+k_2} \end{bmatrix}^\top}_{\mathbf{B}} \underbrace{\begin{bmatrix} \epsilon_1 \\ \vdots \\ \epsilon_p \\ 0 \\ \vdots \\ 0 \\ \epsilon_{p+k_1+1} \\ \vdots \\ \epsilon_q \end{bmatrix}}_{\epsilon}, \quad b_0 \in \mathbb{R}, \mathbf{B} \in \mathbb{R}^q, \epsilon \in [-1,1]^q$$

Now the Hadamard product can be written as

$$\hat{y}' = \hat{x} \odot \hat{z} = \underbrace{a_0 b_0}_{c_0'} + \underbrace{(a_0 \mathbf{B} + b_0 \mathbf{A})}_{C'} \epsilon + \underbrace{\epsilon^\top \mathbf{A} \mathbf{B}^\top \epsilon}_{(*)} \tag{23}$$

$$(*) = \sum_i a_i b_i \underbrace{\epsilon_i^2}_{\in [0,1]} + \sum_i \sum_{j=i+1}^q (a_i b_j + a_j b_i) \underbrace{\epsilon_i \epsilon_j}_{\in [-1,1]} \tag{24}$$

To bring $\hat{y}'$ into zonotope form $\hat{y}$, the term 24 has to be approximated by adding a new error term $\epsilon_{q+1}$ with the error coefficient $c_{q+1}$ and a constant $c_0''$:

$$c_{q+1} = \frac{1}{2}\sum_i |a_i b_i| + \sum_i \sum_{j=i+1}^q |a_i b_j + a_j b_i| \tag{25}$$

$$c_0'' = \frac{1}{2}\sum_i a_i b_i \tag{26}$$

$$\hat{y} = \underbrace{(c_0' + c_0'')}_{c_0} + \underbrace{\left[\; \mathbf{C}' \quad c_{q+1} \;\right]}_{\mathbf{C}} \underbrace{\begin{bmatrix} \epsilon \\ \epsilon_{q+1} \end{bmatrix}}_{\epsilon_{new}}, \tag{27}$$

Unfortunately evaluating equation 25 and 26 is quadratic in the number of error terms in time and when using a matrix formulation utilizing GPU vector operations in space. When the number of error terms is too high and using the transformer described above becomes infeasible, a switch to the box transformer is possible:

$$y_{lb} = \min(x_{lb}z_{lb},\; x_{lu}z_{lb},\; x_{ub}z_{lb},\; x_{ub}z_{ub}) \tag{28}$$
$$y_{ub} = \max(x_{lb}z_{lb},\; x_{lu}z_{lb},\; x_{ub}z_{lb},\; x_{ub}z_{ub}) \tag{29}$$
$$\hat{y} = \frac{y_{ub} + y_{lb}}{2} + \frac{y_{ub} - y_{lb}}{2}\epsilon_{new}, \quad \epsilon_{new} \in [-1,1]^1 \tag{30}$$

### H.6 ENTROPY TRANSFORMER

Based on these elementary transformers, the entropy transformer can be assembled by chaining transformers for the individual component functions according to equation 5, which is reproduced below for convenience.

$$
\begin{aligned}
H(\boldsymbol{y}) &= -\sum_j \frac{e^{y_j}}{\sum_i e^{y_i}} \log\left(\frac{e^{y_j}}{\sum_i e^{y_i}}\right) \\
&= -\sum_j \frac{e^{y_j}}{\sum_i e^{y_i}}\left(y_j - \log\left(\sum_i e^{y_i}\right)\right) \\
&= \log\left(\sum_i e^{y_i}\right) - \sum_j y_j \frac{e^{y_j}}{\sum_i e^{y_i}} \\
&= \log\left(\sum_i e^{y_i}\right) - \sum_j y_j \exp\left(y_j - \log\left(\sum_i e^{y_i}\right)\right) \\
&= c + \log\left(\sum_i e^{y_i-c}\right) - \sum_j y_j \exp\left(y_j - c - \log\left(\sum_i e^{y_i-c}\right)\right)
\end{aligned}
$$

The second term requires four transformers (product, exponential, logarithmic, exponential) adding $3\,n_{class} + 1$ error terms ($n_{class}, n_{class}, 1, n_{class}$) to the output. Since the log-sum-exp term has to be computed only once the first term does not add any additional error terms, while still increasing the corresponding error coefficients.

#### H.6.1 TIGHTNESS

The tightness of the entropy transformer was evaluated in comparison to a box transformer and an upper bound obtained from an optimization approach. To compute the looseness, random input zonotopes with fully populated error coefficient matrices drawn from $\mathcal{N}(0, \sigma_\epsilon^2)$ and centre coefficients drawn from $\mathcal{N}(1, 3^2)$ were created and then propagated through an entropy transformer, before the looseness was computed as the difference between upper and lower bounds. When an input box was required, the zonotopes where converted to a box representation with the same bounds. The mean over 50 samples is reported. The following five different transformer versions were considered:

- *ZonoIter* – The zonotope transformer obtained by chaining the previously introduced transformers and optimizing the slopes $\lambda$ of all transformers sample-wise to minimize looseness.

- *ZonoIterLM* – As ZonoIter, but using the box transformer for the product to obtain a low memory requirement transformer.

- *Zono* – As ZonoIter, but using minimum area slopes instead.

- *ZonoLM* – As Zono, ut using the box transformer for the product to obtain a low memory requirement transformer.

- *Box* – The transformer obtained by using interval arithmetic to propagate bounds.

An analysis of the looseness over input error size, illustrated in figure 12. shows that while the box transformer is clearly the least precise over the whole domain, different errors dominated the behaviour of the various zonotope transformers in different regimes.

The high looseness of Zono and ZonoLM at large input errors suggest that minimum area slopes are not ideal in this regime, while the small penalty for switching to the low memory versions indicates that the error terms incurred from the product transformer are small by comparison to the log and exp contributions. Reducing the input errors on step, flips this behaviour. The product error dominates the differences between minimum area and optimized slopes by a significant margin.

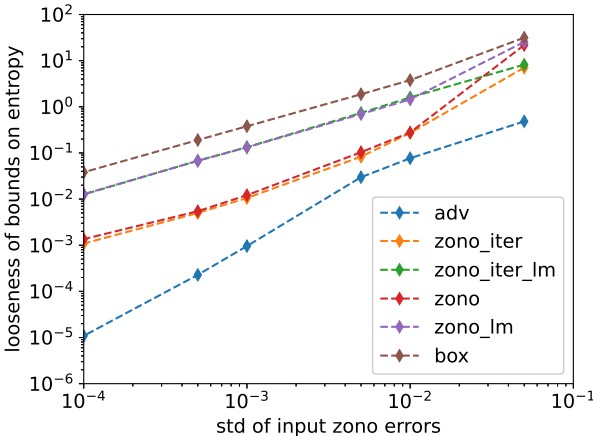

Figure 12: Comparison of the looseness of various versions of the entropy transformer over the standard deviation $\sigma_\epsilon$ of the entries of the input zonotope error coefficient matrix drawn from the distribution $\mathcal{N}(0, \sigma_\epsilon^2)$. *Adv* is a lower bound to the optimal looseness obtained by adversarially attacking the input region, described by the input zonotope.

