# OpenReview forum: "Certify or Predict: Boosting Certified Robustness with Compositional Architectures"
_ICLR.cc/2021/Conference — ICLR 2021 Poster_

### Official Review · AnonReviewer4 · 2020-10-28
**Novel idea with good results that would improved by better experimental evidence and clarity**

**Rating:** 6
**Confidence:** 5

**Review:**

# Summary of Contributions
The paper presents an approach to trade off natural accuracy and certified robustness by combining a network with high natural accuracy (the “core network”) with a second network with high certifiable robustness (the “certification network”). A selection mechanism is used to decide which network an input sample should be processed by. The selection mechanism allows the combined system to perform significantly better than a weighted average of the core and certification networks (e.g. randomly assigning input to the core network with some probability $p$) would.

# Score Recommendation
Despite the weaknesses in experimental evidence, clarity and reproducibility identified below, I recommend an acceptance because the authors have demonstrated that the selection mechanism presented works for non-trivial problems, providing a simple way to trade off natural accuracy and certified robustness.

ACE can consistently benefit from advances that improve the natural accuracy of the core network. In addition, as long as a selection mechanism can be found that is compatible with the certification network, it would be possible for ACE to leverage improvements in certified defenses.

While results are only presented for $l_\infty$ perturbations, I expect that the same approach can be applied to different perturbations, as long as it is possible for the selection mechanism to have a tunable selection rate while having non-trivial robustness to the perturbation of choice.

# Weaknesses
## Missing Experimental Evidence
- The paper claims in the abstract that it is the first to obtain a high natural accuracy with non-trivial certified robustness; the results (91.6% natural accuracy, 22.8% certified robustness) are compared to the prior state-of-the art (77.4% natural accuracy, 16.5% certified robustness). However, I am concerned that the comparison may not be completely fair (if the network the comparison is made to is one of the CROWN-IBP trained DM-Large networks; see the section on “Clarity” below). The paper itself acknowledges (in the first paragraph of page 7) that they did not conduct the extensive training required to obtain the performance reported in the CROWN-IBP paper.
  - The authors should either train the CROWN-IBP DM-Large network with at least as much resources as in the original paper, or clearly identify in both the abstract and introduction that the amount of compute used was constrained.

- The paper claims that “ACE produces much more favorable robustness-accuracy trade-offs than varying hyperparameters of the existing certified defenses” on the basis of comparing DM-Large CROWN-IBP to the Conv2 COLT-based ACE SelectionNet. I believe that more evidence is required to substantiate this claim, since the choice of a base network is rather arbitrary (why choose one comparable to the second DM-Large network, not the first or the third?).
  - One possible set of experiments is to use each of the 5 DM-Large networks with non-trivial certifiable accuracy as the certification network, and then showing that the resulting families of  ACE SelectionNets have a better robustness-accuracy tradeoff.

## Possible Experimental Errors
The Conv3-COLT network in Figure 2 has a performance (~75% natural accuracy, ~50% certifiable accuracy) that is significantly worse than that reported in the original COLT paper (78.4% natural accuracy, 60.5% certified robustness). What is the cause of this significant gap?

## Clarity
- The term “ConvMedBig” is used in the caption for Figure 2 and elsewhere in the paper, but is not defined in the original paper. (It appears that the authors may be referencing [the name in code](https://github.com/eth-sri/colt/blob/20f30b073558ae80e5e726515998c1f31d48b6c6/code/networks.py#L79)). The authors should provide more detail about specifically which network this is. In fact, it appears from the third paragraph of Section 7 (and the code linked above) that “ConvMedBig” matches the network Conv3 exactly. If this is the case, the same name should be used.

- The authors compare to a prior approach with 77.4% accuracy and 16.5% certified robustness but do not specify what this approach is. (It appears from Figure 2 that this may be one of the DM-Large CROWN-IBP networks)

- At the bottom of page 6, the paper states that “the smaller networks to which COLT scales lack capacity to obtain the kind of robustness-accuracy trade-off that we target”. What does this mean? A significant proportion of the results in Table 1 are presented for COLT, so I’m confused by this statement.

## Reproducibility
- Hyperparameters for the PGD attacks used are not provided, making it difficult to understand the strength of the adversarial attack being used. (If the adversarial attack is weak, the adversarial accuracy presented in Table 1 may be significantly higher than the actual robust model accuracy).
- More details should be provided about the algorithm used for certifying the networks in Table 1 (other than the Entropy-COLT-Conv2 network). The third paragraph of Section 5 states that “we only use … convex relaxation-based certification methods based on intervals and zonotopes”, but I couldn’t find any further details (for example, what zonotopes were used to verify the certification network?)

# Questions for Authors
- I’d like to better understand how the adversarial accuracy of the network was evaluated; the paper only mentions that it is “usually computed using an adversarial attack such as PGD” (see the first paragraph of Page 3). One of my concerns is that the selection mechanism (particularly where a selection network is used) may reduce the success of PGD adversarial attacks without increasing the robustness of the network, possibly via gradient masking [1].
- In paragraph 2 of Section 5, the paper specifies that an adversarially trained network was used as the core network. Given that the last paragraph of Section 4 states that “we assume that certification of the core-network always fails”, why did you choose an adversarially trained network (which presumably has slightly worse natural accuracy?)

[1]: Papernot, Nicolas, Patrick McDaniel, Ian Goodfellow, Somesh Jha, Z. Berkay Celik, and Ananthram Swami. "Practical black-box attacks against machine learning." In Proceedings of the 2017 ACM on Asia conference on computer and communications security, pp. 506-519. 2017.


# Additional Feedback
- hyperparamters → hyperparameters (4th line, third paragraph of section 5)
- Figure 2 presents many different networks but it is not clear what the point of the figure is. Compounding the issue is the fact that the corresponding discussion begins almost an entire page later. For improved clarity, the authors should consider adding more to the caption for Figure 2 or moving it closer to the discussion.
- The term “natural accuracy” and “standard accuracy” is used interchangeably; the paper should settle on one.
- Figure 7 labels the y-axis “std of input zono errors” but this term is never introduced anywhere else in the text. (Is this the standard deviation, perhaps?)

# Post-Rebuttal Comments
I've maintained my score at 6.

During the comment period, the authors made progress in improving the clarity of their presentation. As with reviewer 3, I feel that there is still room for improvement; in particular, moving some experiments in Section 5 to the appendix could make for a more focused paper with a clearer message for the reader. (Unfortunately, we did not have enough time during the comment period to get there).

I'd also note that the paper is now at nine pages; this means that I am holding it to a higher bar.

Overall, however, I continue to recommend an acceptance as the method to trade off natural accuracy and certified robustness is simple and significantly improves on the state of the art; for me, these strengths outweight the remaining issues.

---

> ### Author Response · Authors · 2020-11-17
> **Response to Reviewer4 Part I**
>
> We thank the reviewer for their insightful and detailed questions and comments. We did in fact identify some of the same questions but could only conduct the corresponding experiments after the initial submission deadline.
>
> We rephrased your comments and hope to answer all points below:
>
> Q: Why do the CROWN-IBP results you report for DM-Large not match the original results from [4]?
>
> -> In short, we originally used a shorter training schedule and have now rerun the experiments using the full training schedule from [4]. As we believe this question to be important for all reviewers, please refer to the general response where we have answered it in more detail.
>
> Q: Can you provide additional evidence to substantiate the claim that ACE produces a more favourable accuracy-robustness tradeoff than current state-of-the-art methods?
>
> -> We added experiments and visualisations along the lines of Figure 2 and answered this question in more detail in the general response, as we believe it to be important for all reviewers. We did not use all CROWN IBP models to train corresponding ACE models, because we believe it to not be attractive to use networks of the same architecture that are already strictly worse than some ACE models. Instead, we show that even when using significantly weaker certification networks (of a different architecture) there is still a range of natural accuracies, where a corresponding ACE model obtains higher certified accuracies, then the CROWN IBP models (Figure 3 formerly 7).
>
>
> Q: Why do the accuracies you report for Conv3 don’t match those reported by [1]?
>
> -> Note that [1] optimized for a certified accuracy using MILP certification [5] while we instead focus on the much cheaper DeepZ verification [3]. Therefore we trained a network optimized for zonotope certification, which gains around 7.4% of certifiable accuracy at the cost of 4.7% natural accuracy (78.4/40.8) vs (73.7/48.2). We have now added an experiment where we use the network from [1] (and MILP certification for the certification network) as described in the general response (see Figure 2, formerly Appendix C).
>
>
> Q: The Conv3 and ConvMedBig architecture seem to be identical. Why do you use different names?
>
> -> The Conv3 architecture is called ConvMedBig in the repository associated with [1]. We used that name when referring to it as a reference for comparison and used Conv3 when talking about our networks. We agree that this might be confusing, especially since the name ConvMedBig is not used in their paper. We have now homogenized our notation. Note that Conv5 is also the same architecture as DM-Large, where we applied the same rule to decide when to use which name and have now also homogenized our notation to Conv5.
>
>
> Q: Where do you specify how the performance of the state-of-the-art-methods, mentioned in abstract, is obtained? Are they what you measured for the  CROWN-IBP trained Conv5 networks?
>
>
> -> Indeed these numbers were taken from the Conv5 networks we trained with CROWN-IBP.
> We updated the numbers to correspond to the more expensive training schedule discussed above and updated our evaluation section to note where we take these numbers from. We chose the Conv5 setpoint, by first selecting a setpoint for ACE in its intended working range and then selecting the Conv5 setpoint with the largest smaller certified accuracy, to give it the best chance of beating our natural accuracy. We agree that individual setpoints are not ideal for illustrating the trade-off character, therefore we illustrate the trade-off between certified and standard accuracy in Figure 2 (and now also Figure 3, 4, and 7) (formerly 5, 7, and 8)).
>
> EDIT: We rearranged some of our figures for a second revised version and have changed the figure numbers in this response accordingly.
>
> [1] Balunovic, Mislav, and Martin Vechev. "Adversarial training and provable defenses: Bridging the gap." International Conference on Learning Representations. 2019.
>
> [3] Singh, Gagandeep, et al. "Fast and effective robustness certification." Advances in Neural Information Processing Systems 31 (2018): 10802-10813.
>
> [4] Zhang, Huan, et al. "Towards stable and efficient training of verifiably robust neural networks." arXiv preprint arXiv:1906.06316 (2019).
>
> [5] Tjeng, Vincent, Kai Xiao, and Russ Tedrake. "Evaluating robustness of neural networks with mixed integer programming." arXiv preprint arXiv:1711.07356 (2017).

---

> > ### Comment · AnonReviewer4 · 2020-11-24
> > **Response to Author Response, Part I**
> >
> > ## Standardization of Naming
> > Thanks for working to standardize the naming throughout the paper. It looks like there are still a few stray references to DM-Large after it is first introduced in the "Models and Datasets" section of Experimental Evaluation. (See for example the last paragraph on Page 6, the caption of Figure 2, the legend of Figure 2, and the footnotes in Table 1).
> >
> > Also, I'd encourage the authors not to reference the term ConvMedBig, since it's never mentioned in the original paper. Instead, this work could point to the paragraph in https://openreview.net/pdf?id=SJxSDxrKDr where the relevant architecture is described.

---

> > > ### Author Response · Authors · 2020-11-24
> > > **Full Standardization of Naming**
> > >
> > > We thank the reviewer for this note and have now fully standardized the naming and removed any mention of ConvMedBig.

---

> ### Author Response · Authors · 2020-11-17
> **Response to Reviewer4 Part II**
>
> Q: What exactly is meant by the statement that COLT is unsuitable for the accuracy-robustness trade-offs that are target?
>
> -> We train the largest network to which COLT scales (we use the largest network from [1]) using standard training and observe that we are unable to achieve a natural accuracy comparable to what ACE achieves while providing robustness guarantees. Therefore, we conclude that an individual COLT trained network lacks the capacity to achieve high natural (and additionally certifiable) accuracies. The “COLT” entry in Table 1 refers to the method we use to train the provable networks of the ACE architecture (selection- and certification-network) in contrast to using it for an individual provable network. We changed the corresponding sections and hope the new formulation is more clear.
>
>
> Q: How exactly is the adversarial accuracy computed? Could gradient masking have led to an overly optimistic result?
> -> The adversarial accuracy was intended to be purely informative and was not intended as a strong indicator for the true robust accuracy. Therefore, we did not go into more detail on how it is computed. We have now added Appendix E, where we explain this in detail.
> In short: We considered the problem of gradient masking and use the following approach to avoid overly optimistic results. We try to prove that the core- and certification-network can not be reached for a given sample. If this fails, we compute an adversarial attack against the corresponding network in isolation. We are aware that this can lead to samples that would be actually classified correctly by the compositional network, as that specific perturbation might not get selected for classification by the network it successfully attacked.
> We also considered an approach not suffering from this problem (described in Appendix E) but chose to report the more conservative numbers.
>
> Q: Why use an adversarially trained core network?
>
> -> We believe empirical robustness guarantees to also be desirable in domains where deterministic robustness guarantees are required. Therefore we chose to use the more realistic case of using adversarially trained core networks, even though this leads to marginally worse natural accuracies.
>
> Q: Can you give a more detailed description of the certification methods you use?
>
> -> Yes, we use IBP [2] and DeepZ [3] for interval and zonotope certification of IBP and COLT trained networks, respectively. Where explicitly stated, we use MILP certification [5] with some optimizations from [1]. We focus on how we certify the compositional architecture using arbitrary methods instead of how these methods work for individual networks to emphasize the orthogonality of our approach to individual certification methods. We now added additional information in the appropriate section, pointing the reader to the original sources for these methods.
>
> Additional feedback:
> We corrected the typo and homogenized the notation.
> “Std” is indeed short for standard deviation. We adapted the caption to make this more clear.
> We moved Figure 2 to the middle of the corresponding section and hope to have made its intentions more clear in the text preceding it.
>
>
>
> [1] Balunovic, Mislav, and Martin Vechev. "Adversarial training and provable defenses: Bridging the gap." International Conference on Learning Representations. 2019.
>
> [2] Gowal, Sven, et al. "On the effectiveness of interval bound propagation for training verifiably robust models." arXiv preprint arXiv:1810.12715 (2018).
>
> [3] Singh, Gagandeep, et al. "Fast and effective robustness certification." Advances in Neural Information Processing Systems 31 (2018): 10802-10813.
>
> [5] Tjeng, Vincent, Kai Xiao, and Russ Tedrake. "Evaluating robustness of neural networks with mixed integer programming." arXiv preprint arXiv:1711.07356 (2017).

---

> > ### Comment · AnonReviewer4 · 2020-11-24
> > **Response to Author Response, Part II**
> >
> > ## Computing Adversarial Accuracy and Gradient Masking Concerns
> > Thanks for explaining this. Did you include this explanation in the current version of the paper?

---

> > > ### Author Response · Authors · 2020-11-24
> > > **Computing Adversarial Accuracy and Gradient Masking Concerns**
> > >
> > > We include the explanation on how we compute adversarial accuracy in Appendix E and point to it in the corresponding paragraph in the evaluation. We have now also added a short explanation of why we choose an adversarially trained core network.

---

### Official Review · AnonReviewer1 · 2020-10-28
**Combining SOTA network with certified network using an adaptive selection mechanism**

**Rating:** 7
**Confidence:** 3

**Review:**

This paper proposes a new network architecture that combines 1) a state-of-the-art deep neural network with high accuracy (but potentially no robustness certificate), and 2) a small certification network with high certifiable robustness (but not necessarily very high accuracy), using a selection network that adaptively chooses between these two networks. They show that by doing so, the new architecture is able to take advantage of both networks and thus obtain good natural accuracy with better certified robustness that significantly improves upon prior benchmarks.

The main advantage of this framework is its flexibility in allowing arbitrary combinations of STOA deep networks with any networks with certified robustness and their selection mechanism is able to make good use of both.

1. I like this simple idea and I am glad to see its good performance, although I wish the author can develop more theoretical results to quantify the value of a hybrid model.

2. According to (2), the objective may not be differentiable because of the binary function $g$. Can you elaborate on how gradient-based algorithms are applied to this formulation?

3. Can you provide some interpretation of the learned selection mechanism $g_{\theta_s}$? In particular, what features of the samples make them be passed through the core or the certification network?

---

> ### Author Response · Authors · 2020-11-17
> **Response to Reviewer1**
>
> We thank the reviewer for their interesting questions and comments.
>
> We rephrased your comments and hope to answer all points below:
>
> Q: Can you develop more theoretical results to quantify the value of a hybrid model?
>
> ->It is difficult to provide general theoretical guarantees on the certified accuracy of any network, as this depends highly on the underlying distribution. It has for example been shown that distributions can be constructed such that a Bayes optimal classifier with perfect accuracy in the natural case performs arbitrarily badly under arbitrarily small perturbations [1]. Could the reviewer specify what kind of theoretical guarantees they had in mind?
>
> Q: According to (2), the objective may not be differentiable because of the binary function $g$. Can you elaborate on how gradient-based algorithms are applied to this formulation?
>
> -> Yes, indeed using the binary function g directly in a loss function constructed analogously to the compositional network would lead to problems with differentiability. We avoid these issues by leveraging the compositional architecture to decompose the training problem. This allows us to train the certification- and core-networks in isolation, before computing labels for the selection-network and then training it separately as well.
> This turns the training of every individual component-network into a standard (provable) training task and allows us to freely choose training methods independently for every component-network.
>
> Q: Can you provide some interpretation of the learned selection mechanism , in particular, what features of the samples are key for selection?
>
> -> Yes, we can provide some interpretation of how the selection mechanism decides. We observe that if a certification network has difficulty differentiating a group of classes in an adversarial setting (blocks of high off-diagonal terms in the confusion matrix), while they are easy to differentiate from other classes, then classes from this group are selected at a much lower rate for certification. An example for such a group, are the animal classes in CIFAR-10. Whether this effect is due to the selection network learning underlying features that make these samples more difficult to classify provably correct, learning that all of these classes are difficult to certify, or most likely a combination of the two, we can not say with certainty.
> Because the selection-network  was not set up specifically to be an interpretable model (which generally incurs accuracy penalties), it is difficult to pinpoint individual features learned by the selection mechanism.
> We added an experiment comparing three selection networks on an otherwise identical ACE model. We transferred one selection network from a different Conv3 ACE model, trained on using labels based on the adversarial correctness of the sample and trained one in the standard way using provable correctness. We observe that the transferred selection network performs very well, while the one trained using adversarial accuracy performs notably worse. This suggests to us that a) the certification difficulty of a sample is stable over different certification networks at least to some degree and b) that the selection network does learn features distinguishing the difficulty of finding an adversarial example from provable robustness. We present these results in more detail in Appendix D and hope this provides some intuition on how the selection decision is made.
>
> [1] Zhang, Hongyang, et al. "Theoretically principled trade-off between robustness and accuracy." arXiv preprint arXiv:1901.08573 (2019).

---

### Official Review · AnonReviewer3 · 2020-10-29
**Review for Paper3751**

**Rating:** 6
**Confidence:** 5

**Review:**

This paper focuses on improving the standard (clean) accuracy for certifiably
robust models.  To achieve good certified accuracy, previous works typically
make the standard accuracy much worse than naturally trained models. The
authors propose a selection mechanism to choose between a certified model with
low clean accuracy and a naturally trained model with high clean accuracy.  At
a high level, when the certified model cannot certify, there is no point to use
it for classification. A naturally trained model (which cannot be certified as
well) is selected to improve standard accuracy.

Strengths:

Most previous works on certified defense focus on improving certified accuracy,
and standard accuracy is usually sacrificed. This paper focuses on a different
and important setting where high standard accuracy is desired, which is
neglected by many previous works. I think this is a good step.

The proposed selection scheme can balance a certifiably robust model with a
naturally trained but highly accurate model. Such a combination can be helpful
in the settings where high prediction accuracy is required.

The proposed method is technically sound. Using a certified selector makes the
whole network certified when it chooses the certified network. To improve clean
accuracy, The core network is used when the certified selector chooses the core
network (i.e., the selector believes the certified network cannot make a good
prediction on this example) or cannot certify.

The paper overall is well motivated and organized.

Issues and questions:

At a high level, this certification scheme does not improve certified accuracy
(it only makes it worse); it only helps with the verified accuracy vs. clean
accuracy trade-off.  Thus, a crucial part of evaluation is to show the verified
accuracy vs.  clean accuracy tradeoff.  However, it is not well demonstrated in
the experiments. Especially, I think results Table 1 are not so useful because
we can't see how the baseline certified defense models perform and cannot see
this tradeoff. Also, the certified accuracy numbers are really low compared to
other works, and sometimes close to 0 (e.g., on ImageNet-200 only 3% accuracy).
Thus, it is important to show a tradeoff figure here.

I recommend using figures similar to Figure 2 to present the results for all
settings (CIFAR 2/255 and 8/255; downscaled ImageNet-200 at 1/255) (but be
aware Figure 2 has its own issues, see comments below). Importantly, we should
fix a well known certified model (e.g., COLT or CROWN-IBP) and then, apply ACE
with different thresholds to see how the clean accuracy improves with dropped
verified accuracy.  For CIFAR, COLT or CROWN-IBP pretrained models can be used
as the base certified model.  For Imagenet-200, I found a recent work [1]
presented certified defense models on 64*64 TinyImageNet and ImageNet datasets
which can be helpful. They reported around 15% certified accuracy and also uses
much larger model structures which should improve the results in this paper by
using their pretrained models as the base certified model for selection (I
doubt the simple CNN models in this paper are sufficient for ImageNet). Again,
the trade-off part is the most important results to see in this paper, which is
not well demonstrated.

Figure 2 made a misleading comparison because the ACE based methods are using
COLT as the base certified classifier and it is inappropriate to compare it to
CROWN-IBP with different kappas. We should either also use COLT trained with
different weights on natural loss (similar to the kappa in CROWN-IBP) to see
this tradeoff, or use CROWN-IBP as the base certified classifier in this
figure.  Especially, in the CIFAR 2/255 setting, COLT achieves better clean
accuracy than CROWN-IBP, so this gives ACE an advantage in this comparison, and
the claim that ACE achieves a better trade-off than using the tuned kappa
parameters in (CROWN-)IBP training cannot be justified.

It also seems to me that in Figure 2 the CIFAR 2/255 CROWN-IBP numbers are much
worse than the ones reported in CROWN-IBP paper (they reported 28.48% standard
error and 46.03% verified error), but in Figure 2 it is much worse (~35%
standard error and ~50% verified error). If we use the correct CROWN-IBP model,
it should start at a similar place at the ACE based methods in Figure 2, rather
than on the far left. Can you explain?

Conclusion:

I like the aim on standard accuracy and the network selection idea proposed in
this paper, but its current evaluation is partially missing or misleading and
cannot justify all claims. So I cannot recommend accepting its current version.
However I will be glad to discuss with the authors and re-evaluate the paper
based on new evaluation results from the authors. I will be happy to accept this
paper if the authors can address my issues mentioned above.

---
### After rebuttal

See my reply below for my comments after rebuttal. Overall I feel the paper still has room for improvement and there are several open issues, but it has been improved so it is marginally above acceptance threshold now.

---
Reference:

[1] Xu, Kaidi, et al. "Provable, Scalable and Automatic Perturbation Analysis on General Computational Graphs" https://arxiv.org/pdf/2002.12920

---

> ### Author Response · Authors · 2020-11-17
> **Response to Reviewer3**
>
> We thank the reviewer for their insightful and detailed questions and comments. We did in fact identify some of the same questions but could only conduct the corresponding experiments after the initial submission deadline.
>
> We rephrased your comments and hope to answer all points below:
>
> Q: Can you provide Figures along the lines of Figure 2 for all experiments demonstrating the improvement in tradeoff between natural and certified accuracy and add experiments with established certification networks?
>
> -> Yes, we have added Figures 3, 4, and 7 (formerly 5, 7 and 8) in the style of Figure 2 for CIFAR-10 at 2/255, 8/255 and TinyImageNet at 1/255, respectively. As we believe this question to be important for all reviewers, please see a more detailed answer in the general response.
>
> Q: Can you use the strong provable networks trained by [2] to demonstrate good results are attainable using ACE at ImageNet sized tasks?
>
>
> -> Yes, we added results on TinyImageNet at 1/255 using the WideResNet from [2] both as a certification network and as a feature extractor for the selection network. Using these networks, we achieve significantly better results than on ImageNet200, despite not being able to leverage full sized images on the core-network. This demonstrates that ACE scales to more challenging tasks if sufficiently strong, underlying provable networks are available. This experiment is illustrated in Figure 4 (formerly 8).
>
>
> Q: Can you perform an experiment either comparing COLT based ACE models with COLT trained networks using different natural loss components or comparing CROWN-IBP based ACE models to CROWN-IBP networks trained with different $\kappa$?
>
> -> Yes, we have done both. We added an experiment comparing CROWN-IBP trained Conv5 models with different $\kappa$ with ACE models based on some of these Conv5 networks for CIFAR10 at 8/255 (Figure 3 formerly 7) and an experiment comparing COLT trained Conv3 networks with various natural loss components (Note that COLT usually does not use a natural loss component) with COLT based ACE models for CIFAR-10 at 2/255 (Figure 2 formerly 5). However, we don’t believe the latter to be completely fair, as COLT only scales to small models which lack the capacity to achieve a comparable natural accuracy, even when trained without any robustness considerations.
>
> Q: Is it misleading to compare an ACE model based on COLT with individual models using a cheaper certified training method such as CROWN-IBP?
>
> -> No, we believe it to be a feature of the ACE architecture that small models with high certified accuracies, trained and certified with expensive provable training and certification methods, can be combined with larger core-networks to obtain ACE models that achieve high natural accuracies, typically not accessible to networks of a size to which these expensive methods scale. We believe a comparison with more scalable provable training methods to be appropriate if these are required to scale to the larger networks required to achieve comparable natural accuracies.
> However, we want to point out that the certification networks we used for the comparison in Figure 2 are actually worse than the CROWN-IBP networks and that it requires less than half the GPU time to train and certify these models. Using more expensive certification methods such as MILP, which don’t scale to Conv5, we can improve the performance of the ACE model further.
>
>
> EDIT: We rearranged some of our figures for a second revised version and have changed the figure numbers in this response accordingly.
>
>
> [1] Zhang, Huan, et al. "Towards stable and efficient training of verifiably robust neural networks." arXiv preprint arXiv:1906.06316 (2019).
>
> [2] Xu, Kaidi, et al. "Automatic Perturbation Analysis on General Computational Graphs." arXiv preprint arXiv:2002.12920 (2020).
>
> [3] Balunovic, Mislav, and Martin Vechev. "Adversarial training and provable defenses: Bridging the gap." International Conference on Learning Representations. 2019.

---

> > ### Comment · AnonReviewer3 · 2020-11-23
> > **Rating increased based on the new results however there are still some issues**
> >
> > Thank you for providing the additional results. These results are very helpful and essential for this paper.
> >
> > I am mostly convinced that the proposed method can perform better then directly
> > tuning a weight on natural training loss (the "kappa" parameter) in existing
> > works. From Figure 5, 7 and 8 in Appendix it seems the author's claim is
> > supported. So I am increasing my rating by 1.
> >
> > There are still several presentation issues in this paper however, so I cannot
> > give a firm accept for this paper. From Figure 2 and Table 1 in the main text,
> > if I don't read the new results in appendix very carefully, I am still not
> > convinced that the proposed approach is better than directly weighting the loss
> > function. The main concern is that, the issue in Figure 2 is still unresolved
> > in the updated revision: for COLT there is no comparison against simple loss
> > function weighting (I understand the original COLT paper does not use a natural
> > loss, but it can be easily added and tuned for a trade-off for clean accuracy),
> > and for CROWN-IBP there is no ACE variants. Also Table 1 is not really helpful
> > here since from this table only one data point for each setting is shown and it
> > is impossible to show the trade-off. I would suggest using Figure 7, Figure 8
> > to present the results.
> >
> > Additionally, I am also confused why the authors use an adversarially trained
> > network as the core classification network (same question asked by
> > AnonReviewer4). It seems using adversarial trained model provides no benefits
> > for the metrics reported in the paper.  The answer for AnonReviewer4 is not
> > very convincing to me. Is the real reason somewhat related to the adversary
> > accuracy in Table 1?
> >
> > Over all the paper looks little bit rushed and can be confusing for people not
> > very familiar with related papers. For example:
> >
> > 1. In Figure 7, there are several triangles not presented in the legend. Also the green, red lines are CROWN-IBP not IBP (lengend label is wrong).
> >
> > 2. Table 1 also has a similar issue, it seems both CROWN-IBP and (Xu et al.) are referred to as "IBP" under the "Provable training" column, which is inaccurate.
> >
> > 3. Some discussions on the newly added training method (Xu et al.) for TinyImageNet should be added in section 3. Is their training method very different from COLT or CROWN-IBP (any reason why COLT and CROWN-IBP do not work on TinyImageNet)?
> >
> > 4. In Figure 2 the "Conv2, COLT" marks look like black for me, not gray (as mentioned in text). Also I think there are too many lines on this figure; I think just showing one setting of COLT is sufficient here (e.g., Conv 3 COLT).
> >
> > Thanks again for the response and I hope my suggestions above can help the authors further improve their paper.

---

> > > ### Comment · AnonReviewer4 · 2020-11-24
> > > **Agree that presentation of paper could be improved**
> > >
> > > I'd like to echo Reviewer 3's sentiment that the presentation of the paper could be improved. In particular, for me, it's still hard to figure out what the takeaway from Figure 2 and Table 1 is - even after reading the surrounding text.
> > >
> > > ### Figure 2
> > >
> > > Here are some possible takeaways from Figure 2 as it currently stands. Which of these should readers focus on?
> > >
> > > - For a range of different _types_ of core networks (colored triangles), ACE SelectionNet is able to provide a good certifiable accuracy - natural accuracy tradeoff (colored squares)
> > > - ACE SelectionNet provides a better certifiable accuracy - natural accuracy tradeoff than CROWN-IBP (at a particular value of certifiable accuracy or natural accuracy, the yellow gradient is steeper than the other colors)
> > > - ACE Entropy has better performance than the ACE SelectionNet (comparing the black diamond and the black square)
> > >
> > > As Reviewer 3 mentions, showing just one setting of COLT along with the associated selection net would suffice here if the intention is to compare the certifiable accuracy - natural accuracy tradeoff. In any case, it would be helpful to have a concise guide to the Figure in the caption.
> > >
> > > ### Table 1
> > >
> > > I'm not sure what I should be taking away from this table.
> > >
> > > - Is it simply meant to present results for a range of datasets and $\epsilon$ values? (If so, one result per dataset / $\epsilon$ pair should suffice).

---

> > > > ### Author Response · Authors · 2020-11-24
> > > > **Rearranged presentation of results**
> > > >
> > > > We thank the reviewer for their suggestions and have rearranged our results to improve the presentation. We have again summarized the main points and answer them below:
> > > >
> > > >
> > > > Q: Can you point the reader towards what he should focus on in Figure 2?
> > > >
> > > > -> Following your and Reviwer3’s comments, we split the old Figure 2 in the new Figure 2 and 7, and provide a concise note on what to focus on in the caption. The intent of the new Figure 2 is twofold: (i) Show that using the same architecture and training method, varying the natural loss component yields significantly lower certified accuracies at a given natural accuracy than using an ACE model. (ii) Point out that using the same architecture and training method, but a more expensive certification method, the ACE performance increases in line with the individual network performance.
> > > >
> > > > Q: Is the intent of Table 1 to simply present results on a range of datasets and perturbation sizes?
> > > >
> > > > -> Yes, the intent of Table 1 was to show that ACE can achieve consistent results over a range of datasets, architectures and perturbation sizes. We have reduced its length significantly, moving a full copy to Appendix F.

---

> > > ### Author Response · Authors · 2020-11-24
> > > **Response to second comment of Reviewer3**
> > >
> > > We thank the reviewer for their quick response, allowing us to address the remaining points as well. We have again summarized the main points and answer them below:
> > >
> > > Q: Can you provide a comparison of a COLT based ACE model with a COLT model using loss function weighting in Figure 2?
> > >
> > > -> Yes, we have changed the organization of our results and now present a comparison of three COLT based ACE models with individual COLT networks trained using different natural loss components in Figure 2. We have moved the comparison to the CROWN-IBP trained models to Appendix C Figure 7.
> > >
> > > Q: Can you provide an ACE model using a CROWN-IBP trained network as a certification network?
> > >
> > > -> Yes, we reorganized our results for CIFAR-10 at 8/255 slightly and moved them to the main paper body to improve clarity. We trained 3 ACE models using 2 different CROWN-IBP trained Conv5, and one IBP trained Conv3 network as certification-networks. We trained all selection-networks using IBP training. We compare these ACE models with individual, CROWN-IBP trained Conv5 networks in Figure 3. We use IBP instead of CROWN-IBP to train the selection network to simplify the implementation and changing this would only improve the ACE models, so we believe the conclusions we draw from the comparison are valid. We updated the experimental evaluation section to reflect these changes and improve clarity.
> > >
> > > Q: Can you use Figure 7 and 8 instead of Table 1 to present in the main body of the paper?
> > >
> > > -> Yes, we have pulled Figure 7 and 8 forward into the main body and reduced Table 1, providing an extended version in Appendix E. The old Figure 7 corresponds to the new Figure 3 and Figure 8 to Figure 4.
> > >
> > > Q: Why are you using an adversarially trained core-network?
> > >
> > > -> As stated in the reply to Reviewer4, we did want to provide a fair comparison representative of potential real world applications, which would almost certainly use adversarially trained core-networks. Indeed we could use normally trained networks to improve natural accuracy at the cost of adversarial accuracy. We added this explanation to the corresponding paragraph and note that we report the most conservative adversarial accuracy numbers obtained with the methods described in Appendix F.
> > >
> > > Q: Why are several triangles in Figure 7 (now 2 and 3) not presented in the legend?
> > >
> > > -> The triangles you are referring to, were intentionally omitted to avoid an overcrowded legend. They represent the certification-networks corresponding to the identically colored ACE models. We have updated the caption to make this more clear.
> > >
> > > Q: Why are both the networks using  CROWN-IBP and LiRPA trained certification networks marked as IBP trained?
> > >
> > > -> We train the full selection-network in the first case and its last layer in the second case using IBP. We provide footnotes in both cases in Table 1 and added clarifications to the captions of all affected figures as well as the corresponding paragraphs in the experimental evaluation section, to clarify this point.
> > >
> > > Q: How is LiRPA different from COLT and (CROWN-)IBP? Can you add a discussion of the LiRPA training method in Section 3?
> > >
> > > -> LiRPA is based on linearly bounding all activations and outputs in a very similar manner to CROWN-IBP, differentiating itself mostly by the use of loss fusion. [3] introduce loss fusion as the concept of directly upper bounding the robust cross-entropy loss instead of lower bounding the margin between the network output associated with the ground truth label and all others and computing a cross entropy loss from these margins. This leads to slightly tighter bounds and improves scaling to tasks with many classes such as TinyImageNet significantly. Furthermore, its implementation permits more complex network architectures to be considered. COLT has a completely different approach, computing adversarial examples in the latent spaces of the original input region and conducting adversarial training with these. We added LiRPA more explicitly to the related work section and clarified in section 3 that in addition to using IBP and COLT for training, we use networks pretrained with CROWN-IBP and LiRPA.
> > >
> > > Q: Can you improve the clarity of Figure 2 as it currently looks quite crowded?
> > >
> > > -> Yes, to improve the clarity we split Figure 2 in the new Figure 2 and Figure 7, removed the Conv2 results, refer to the very dark grey as black from now on and added a note regarding what to focus on to the caption.
> > >
> > > [1] Balunovic, Mislav, and Martin Vechev. "Adversarial training and provable defenses: Bridging the gap." International Conference on Learning Representations. 2019.
> > > [2] Zhang, Huan, et al. "Towards stable and efficient training of verifiably robust neural networks." arXiv preprint arXiv:1906.06316 (2019).
> > > [3] Xu, Kaidi, et al. "Automatic Perturbation Analysis on General Computational Graphs." arXiv preprint arXiv:2002.12920 (2020).

---

> > > > ### Comment · AnonReviewer3 · 2020-11-25
> > > > **Thank you for the response. Please make sure to take more time to polish the paper and address the issues below in final version.**
> > > >
> > > > Thank you for the response, and the results are presented in a better way now in the last revision. Only some issues left:
> > > >
> > > > 1. In Figure 7 (now in appendix) the CROWN-IBP (Conv5) + ACE results seem still missing, so this figure is not consistent as Figure 2 and 3. In each of the figure, we should have both ACE models and existing models (using the same certification network) under different weights (kappa) of natural loss. Please make sure to add it in the final revision.
> > > >
> > > > 2. In Table 1 and 4, it can be more clear if two columns are presented, one showing the training method of certification network (COLT/CROWN-IBP/LiRPA) and one showing the ACE training method (COLT/IBP).
> > > >
> > > > 3. Despite the arguments made by the reviewer, I think it is still better to add some results using naturally trained models at least for one setting (e.g., CIFAR eps=8/255), because future works may use this paper as a baseline and using naturally trained models is an important setting.
> > > >
> > > > 4. The purple triangle on Figure 4 is missing.
> > > >
> > > > Thank you for explaining to me the difference between COLT/CROWN-IBP and (Xu et al.). It is now clear to me.
> > > >
> > > > Based on the new revision, I have increased to score to 6 (assuming the authors will address the remaining issues above). I believe the paper still has room for improvements so cannot further increase my score. Please make sure to take more time to polish the paper and address the issues above.

---

> > > > > ### Author Response · Authors · 2020-11-25
> > > > > **Response to final comments by Reviewer3**
> > > > >
> > > > > We thank the reviewer for their valuable suggestions. We updated the paper and answered the questions below:
> > > > >
> > > > > Q: Can you add results for CROWN-IBP trained ACE networks to the comparison at CIFAR-10 at 2/255?
> > > > >
> > > > > -> Yes, we will address this in the next version and expect results in line or slightly better than those observed for the DeepZ certified, COLT trained networks.
> > > > >
> > > > > Q: Can you add a clear distinction between the method used to train the certification- and the selection-network to Tables 1 and 4?
> > > > >
> > > > > -> Yes, we have updated the mentioned tables accordingly, adding an extra column.
> > > > >
> > > > > Q: Can you provide results for ACE models, with normally trained core-networks?
> > > > >
> > > > > -> Yes, we have added such results in Appendix F Figure 9 and will add a more thorough comparison for the next version.
> > > > >
> > > > > Q: Can you add the missing purple triangle in Figure 4?
> > > > >
> > > > > -> Yes, we have done so.

---

### Author Response · Authors · 2020-11-17
**General Answers**

We thank reviewers for their feedback and comments. We first list some general updates we have made to the submission, then proceed to answer common questions in the reviews. Finally, we respond to questions of each individual reviewer.

General updates:
We updated our IBP certification implementation to obtain tighter bounds on the deltas between output activations, improving the certified accuracies for our IBP certified networks. This does not influence the baseline networks, as we did not use our implementation for their certification.
As the architectures Conv3 and Conv5 are identical to the 3 convolutional layer architecture from [2] and DM-Large from [1] we have changed the notation to the unified ConvX standard.

Common questions:

Q: Why are the accuracies for the CROWN-IBP trained DM-Large network notably worse than the original results from [1]?


-> As multi GPU training in [1] takes a long time, we originally used their single GPU training schedule provided by [1] to have results ready by the submission deadline (thus indeed achieving worse results). Since then, we have reran the training for all DM-Large(=Conv5) models using the multi GPU training schedule suggested by [1] and have obtained results very similar to what they report. We have updated all our figures and evaluation accordingly. Despite both of our certification networks now being inferior to some of the CROWN-IBP Conv5 networks, we still achieve significantly higher certified accuracy at high standard accuracies, highlighting the efficacy of our method.


Q: Can you provide additional evidence to substantiate the claim that ACE produces more favourable accuracy-robustness tradeoff than current state-of-the-art methods?


-> Yes, we agree that Table 1 is not an ideal way to present results on the tradeoff and have added Figures 3, 4, and 7 (formerly 5, 7 and 8) in the style of Figure 2 for CIFAR-10 at 2/255, 8/255 and TinyImageNet at 1/255, respectively. For CIFAR-10 at both perturbation levels we trained multiple CROWN-IBP networks at different tradeoffs as a reference. For TinyImageNet this was unfortunately not feasible in the given timeframe. For CIFAR-10 at 2/255 we use the Conv3 directly from [2] as a certification-network for our ACE model and use MILP [4] to certify the certification-network. We compare with models trained using COLT with different natural loss components, but can only certify them using DeepZ [3], as MILP certification would take in excess of a week per setpoint. Therefore, we also evaluate our ACE models using DeepZ and report results in Figure 2 (formerly 5). For CIFAR10 at 8/255 we use one Conv5 model directly from [1] and one trained ourselves at a higher natural loss component as certification-networks for ACE models. Additionally we add weaker Conv3 models to the comparison and show that for all ACE models there are setpoints where they outperform the individual Conv5 models. This is illustrated in Figure 3 (formerly 7).

EDIT: We rearranged some of our figures for a second revised version and have changed the figure numbers in this response accordingly.

[1] Zhang, Huan, et al. "Towards stable and efficient training of verifiably robust neural networks." arXiv preprint arXiv:1906.06316 (2019).

[2] Balunovic, Mislav, and Martin Vechev. "Adversarial training and provable defenses: Bridging the gap." International Conference on Learning Representations. 2019.

[3] Singh, Gagandeep, et al. "Fast and effective robustness certification." Advances in Neural Information Processing Systems 31 (2018): 10802-10813.

[4] Tjeng, Vincent, Kai Xiao, and Russ Tedrake. "Evaluating robustness of neural networks with mixed integer programming." arXiv preprint arXiv:1711.07356 (2017).

---

> ### Comment · AnonReviewer4 · 2020-11-24
> **Apparent Inconsistenct between Abstract and Introduction**
>
> With the updated results on CROWN-IBP, it looks like the abstract was updated but the introduction was not. Quoting the paper as at the time of this comment:
>
> ### Abstract
>
> > For instance, on CIFAR-10 with an `$l_∞$ perturbation of 2/255, we are the first to obtain a high natural accuracy (90.1%) with non-trivial certified robustness (27.5%). Notably, prior state-of-the-art methods incur a substantial drop in accuracy (79.5%) for a similar certified robustness (24.6%).
>
> ### Introduction, third paragraph
>
> >  For example, on the challenging CIFAR10 dataset with an `∞ perturbation of 2/255, we obtain 91.6% natural accuracy and a certified robustness of 22.8%. On the same task, prior approaches cannot obtain the same natural accuracies
> for any non-trivial certified robustness and, when tuned to comparable levels of certified robustness (16.5%), only obtain a natural accuracy of 77.4%

---

> > ### Author Response · Authors · 2020-11-24
> > **Resolving the inconsistency between abstract and Introduction**
> >
> > We thank the reviewer for this observation. We have resolved the inconsistency as described in the reply above.

---

> ### Comment · AnonReviewer4 · 2020-11-24
> **Network selected for comparison in the abstract**
>
> In the abstract, the paper states that
>
> > we are the first to obtain a high natural accuracy (90.1%) with non-trivial certified robustness (27.5%). Notably, prior state-of-the-art methods incur a substantial drop in accuracy (79.5%) for a similar certified robustness (24.6%)
>
> (It appears that the comparison is being made with the "DM-Large, CROWN IBP" network (yellow) that is third from the right.)
>
> This gives the impression that ACE produces networks with higher natural accuracy AND certified robustness than the state of the art, while in fact ACE explicitly trades off some certified robustness for a higher natural accuracy.
>
> It seems like the fairer comparison here would be with the "Conv3, COLT & MILP" network. This network has only a slightly higher drop in accuracy (to `~78%`) but with a far higher certified robustness (`~60%`). (In fact, I expect that it would be possible to train a slightly larger "COLT & MILP" network that has a natural accuracy greater than the 79.5% value above).

---

> > ### Author Response · Authors · 2020-11-24
> > **Comparison with state-of-the-art in the abstract and introduction**
> >
> > We thank the reviewer for their quick response, allowing us to address the remaining points as well. We have again summarized the main points and answer them below:
> >
> > Q: Why don’t you compare your results to “Conv3, COLT_MILP” which achieves a much higher certified robustness in the abstract?
> >
> > -> We originally did not reference any results computed with MILP as we did not want to focus on numbers where the very high computational requirements (up to an hour per sample) make the computation of a tradeoff curve for multiple reference models infeasible. This is also the reason why we did not report the MILP certified numbers of the COLT ACE model in the abstract and introduction. We decided to remove the reference numbers altogether, as we can not provide sufficient context to put them into the right perspective. We note that while it might very well be possible to train a larger COLT_MILP model the certification of even a Conv3 model already takes well above a week for the full test set.

---

### Decision · Program_Chairs · 2021-01-07
**Final Decision**

**Decision:**

Accept (Poster)

**Comment:**

This paper proposes a selection mechanism to choose between a certified model with low clean accuracy and a naturally trained model with high accuracy, to improve the standard clean accuracy for certifiably robust models.  At a high-level, the idea behind this combined system is that when the certified model cannot certify, one should avoid using it for classification, but rather should use a naturally trained model.  A state-of-the-art naturally trained networks is used as the "core network", and a small certification network with high certifiable robustness is used as the "certification network". The major contribution is a selection network that adaptively chooses between these two networks.

Pro
+ The idea of using two networks adaptively is novel. The proposed selection mechanism has been shown to be able to combine the merits of both networks to obtain better natural accuracy with good certified robustness.


Con
- The experiment section still has room for improvement. Specifically, the presentation of the results were not convincingly conveying the tradeoff between the clean accuracy and the certified accuracy.  After the rebuttal, the authors made some improvements that addressed many of the concerns about the clarity and reproducibility issues. However, reviewers suggest further polishing the experiment section.

Overall, I think the novelty of the paper combined with the promising results achieved outweigh the presentation issues. I would recommend accepting this paper.